# Runtime Analysis of Evolutionary NAS for Multiclass Classification

Zeqiong Lv [1]   Chao Qian [2]   Yun Liu [1]   Jiahao Fan [1]   Yanan Sun [✉ 1]

## Abstract

Evolutionary neural architecture search (ENAS) is a key part of evolutionary machine learning, which commonly utilizes evolutionary algorithms (EAs) to automatically design high-performing deep neural architectures. During past years, various ENAS methods have been proposed with exceptional performance. However, the theory research of ENAS is still in the infant. In this work, we step for the runtime analysis, which is an essential theory aspect of EAs, of ENAS upon multiclass classification problems. Specifically, we first propose a benchmark to lay the groundwork for the analysis. Furthermore, we design a two-level search space, making it suitable for multiclass classification problems and consistent with the common settings of ENAS. Based on both designs, we consider (1+1)-ENAS algorithms with one-bit and bit-wise mutations, and analyze their upper and lower bounds on the expected runtime. We prove that the algorithm using both mutations can find the optimum with the expected runtime upper bound of $O(rM \ln rM)$ and lower bound of $\Omega(rM \ln M)$. This suggests that a simple one-bit mutation may be greatly considered, given that most state-of-the-art ENAS methods are laboriously designed with the bit-wise mutation. Empirical studies also support our theoretical proof.

## 1. Introduction

Neural architecture search (NAS) can automate the design of effective deep neural architectures (Elsken et al., 2019), making itself a crucial step in automating machine learning (He et al., 2021). Evolutionary NAS (ENAS) (Real et al., 2017; Liu et al., 2018; Real et al., 2019; Ünal & Başçiftçi,

2022; Liu et al., 2023; Miikkulainen et al., 2024) employs evolutionary techniques, primarily evolutionary algorithms (EAs), for the automation and has been widely used in real-world applications (Sun et al., 2019b; So et al., 2021; Liu et al., 2023; Yang et al., 2023; Yan et al., 2024). However, the theoretical research of ENAS is still underdeveloped.

Generally, the theoretical research of ENAS follows the conventional theory of EAs. One essential topic in this aspect is the runtime analysis (Auger & Doerr, 2011; Neumann & Witt, 2010; Zhou et al., 2019; Doerr & Neumann, 2020), which represents the expected number of fitness evaluations until an optimal or approximate solution is found for the first time. In practice, the runtime analysis is very challenging, primarily due to the randomized nature of EAs. Therefore, the community typically begins with (1+1)-EA with various mutations upon benchmark problems which have mathematically formulated fitness functions. For example, theoreticians (Droste et al., 2002; Doerr et al., 2008; Doerr & Goldberg, 2013; Witt, 2013) have analyzed the runtime of (1+1)-EA with one-bit mutation or bit-wise mutation on ONEMAX and LEADINGONES functions. These analyses further provide theoretical insights for configuring EAs to solve complex problems like the dynamic makespan scheduling problem, where (1+1)-EA with bit-wise mutation efficiently maintains a good discrepancy when the processing time of a job changes dynamically (Neumann & Witt, 2015).

In the context of runtime analysis, mathematically formulated fitness function plays the role of directly assessing the progress of the search process of EAs (Auger & Doerr, 2011; Neumann & Witt, 2010; Zhou et al., 2019; Doerr & Neumann, 2020). However, constructing such a mathematically formulated fitness function for ENAS is very challenging. This is because the fitness of a neural architecture is obtained by training itself with learning algorithms, of which the whole process is difficult to be mathematically formulated since it is inherently black-box.

In the literature, there is only rare work making attempts to mathematically formulate fitness functions for machine learning tasks. Fischer et al. (2023) considered the classification of points on the unit hypersphere and constructed a fitness function that quantitatively reflects the proportion of correctly classified points. This is done by leveraging

[1]College of Computer Science, Sichuan University, China [2]National Key Laboratory for Novel Software Technology, and School of Artificial Intelligence, Nanjing University, China. Correspondence to: Yanan Sun <ysun@scu.edu.cn>.

*Proceedings of the 42nd International Conference on Machine Learning*, Vancouver, Canada. PMLR 267, 2025. Copyright 2025 by the author(s).

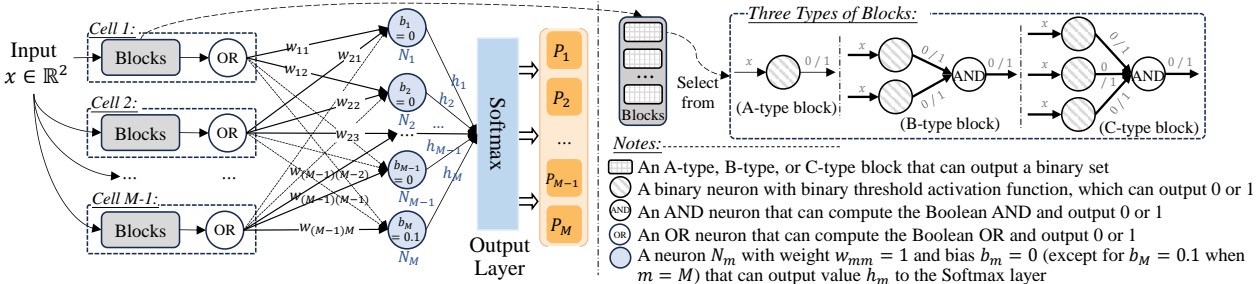

*Figure 1.* The neural architecture skeleton, including a set of cells where each of them consists of a set of blocks. There are three types of blocks: A-type, B-type, and C-type. Each block receives the input $x$ and outputs a binary bit (1 or 0). Thus, $M - 1$ cells will output a binary set for each neuron in $\{N_1, N_2, \ldots, N_M\}$. Each neuron $N_m$ computes a value $h_m$, which is then normalized by the Softmax layer to yield the probability $P_m$. The class label corresponding to the highest probability is chosen as the final classification result.

the hyperplane parameters representing a neural network, thus providing a way to evaluate the neural networks. Upon this, Fischer et al. (2024) introduced a bending hyperplane and used its parameters to mathematically formulate a more advanced fitness function. However, the fitness functions in these works evaluate neural networks with a fixed architecture, which is in contrast to ENAS, where the architectures are varied through evolution. To this end, Lv et al. (2024b) developed an intuitive fitness function for the evolved neural architectures. This function incorporates polytope-based decision boundaries alongside hyperplanes and mathematically quantifies the volume of the correct decision regions. However, this work only focused on binary classification problems. In practice, many real-world applications, such as image recognition, speech recognition, and medical diagnosis, require distinguishing between multiple classes.

On the other hand, the representation of the search space in the current theoretical works also hinders the runtime analysis of ENAS for multiclass classification. So far, only one related work has been conducted (Lv et al., 2024b), but the search space considered has some limitations. Specifically, it is unable to handle multiclass classification problems, since the design of the neural architecture's skeleton only enables distinguishing between two decision spaces corresponding to two classes. Moreover, the search space restricted the analysis of ENAS algorithms with common evolutionary strategies, such as the bit-wise mutation operator, as it resulted in a solution of length three. To start the runtime analysis of ENAS for multiclass classification problems, a more powerful and practical search space is urgently needed.

This work makes an initial attempt for the runtime analysis of the ENAS algorithm for a multiclass classification problem. The main contributions are summarized below:

1) We propose a multiclass classification benchmark prob-

lem MCC with $M$ classes, based on which a fitness function is mathematically formulated to evaluate the quality of any given neural architecture in solving this problem. The fitness function can serve as a benchmark for other theoretical aspects of ENAS, which then can help enhance the understanding of ENAS and provide insight for designing better ENAS algorithms.

2) We design a more practical search space with two inter-related levels, where the first level is cell-based and the second level is block-based. This search space with solution length $M$ is consistent with the common setting of ENAS and supports the theoretical analysis of ENAS for multiclass classification problems.

3) We analyze the expected runtime bounds of (1+1)-ENAS algorithms with one-bit and bit-wise mutations in searching for the optimal neural architecture of MCC. The proven runtime bounds (**Theorems 4.1 to 4.4**) show that one-bit and bit-wise mutations achieve nearly the same performance for (1+1)-ENAS. We also conduct empirical analysis to verify the theoretical proofs. To the best of our knowledge, this is the first theoretical work of ENAS for multiclass classification problems.

## 2. Preliminaries

This section presents the foundations of considered neural architectures for multiclass classification, followed by introducing the ENAS algorithm.

### 2.1. Considered Neural Architectures

We consider a neural architecture for multiclass classification. This architecture reduces the multiclass classification to multiple binary classifications (Friedman et al., 2000), each is solved by a binary classifier. The final classification result is then obtained by aggregating the weighted outcomes from these binary classifiers (Aly, 2005).

The skeleton of the neural architecture is shown in Figure 1. The input instance is fed into $M - 1$ cells, each of which is treated as a binary classifier that outputs either 0 or 1. The outputs of these $M - 1$ cells are then fed into a hidden layer consisting of $M$ neurons. The hidden layer is designed to collect the results from each of the $M - 1$ binary classifiers and integrate them to prepare for computing the probabilities of the input belonging to each class. We fix the biases and weights of the $M$ neurons as follows: 1) except for the last neuron $N_M$, whose bias is $b_M = 0.1$, all other neurons have a bias of 0; 2) the weights satisfy

$$\begin{cases} \forall i : w_{i,i} = 1, w_{i,i-1} = 0.5, w_{i,i+1} = 0.4, \\ \forall j \notin \{i-1, i, i+1\} : w_{i,j} = 0. \end{cases}$$

Such settings make the output of $N_M$ be maximized when the cells output $0^{M-1}$, thereby correctly classifying the $M$-th class. For any output $h_i \in \{h_1, h_2, \ldots, h_M\}$ from the $M$ neurons, the softmax activation function (Sharma et al., 2017) is applied to compute $e^{h_i} / \sum_{j=1}^{M} e^{h_j}$ as the probability $P_i$ of the input instance belonging to class $i$. Finally, the classification result is class $\arg\max_{i \in [1..M]} P_i$.

The core of the considered neural architecture is *cell*, which serves as a fundamental building unit commonly used in the ENAS community (Elsken et al., 2019; Liu et al., 2023; Zoph et al., 2018; Real et al., 2019). Each cell consists of $l$ *blocks* and one OR neuron, allowing for flexible and intricate topological structures (Sun et al., 2019a; Zhong et al., 2018). Each block will output a binary bit. Then, the OR neuron in the cell will receive $l$ bits from the $l$ blocks and output either 0 or 1.

Three types of block are sufficient to build a neural architecture that can tackle most classification problems: A-type, B-type, and C-type blocks, as suggested in (Lv et al., 2024b) and shown in the top-right of Figure 1. Each block contains at least one binary neuron with binary step function (Sharma et al., 2017), and at most one AND neuron to compute the Boolean AND of the binary outputs from the previous neurons. Based on this, these three types of blocks allow the neural architecture with 2-dimensional input to form decision regions (Gibson & Cowan, 2002; Nguyen et al., 2018) in the shape of segment, sector, and triangle, respectively. These decision regions are representatives of half-space, unbounded polyhedron, and bounded polyhedron, and most classification problems can be described as their disjoint union (Bertsimas & Tsitsiklis, 1997). Further details about the considered neural architecture are provided in **Appendix A**.

## 2.2. ENAS Algorithm

ENAS algorithms aim to search for an optimal or approximate neural architecture from a search space consisting of all potential solutions (Elsken et al., 2019; Liu et al., 2023).

As detailed in Section 2.1, a neural architecture consists of $M - 1$ cells, each of which includes A-type, B-type, and C-type blocks. We let the number of these blocks in the $m$-th cell be $n_A^m$, $n_B^m$, and $n_C^m$, respectively. Then, we encode a neural architecture using $M - 1$ triplets of integers, where each triplet corresponds to the counts of each block type in a cell. The encoding of a neural architecture is given by:

$$\boldsymbol{x} = \{(n_A^1, n_B^1, n_C^1), \ldots, (n_A^{M-1}, n_B^{M-1}, n_C^{M-1})\}.$$

Based on this encoding mechanism, the search space is $\mathcal{S} = \{\mathbb{Z} \times \mathbb{Z} \times \mathbb{Z}\}^{M-1}$. This is a two-level search space with cells at the first level and blocks at the second level. By following the convention in EA's runtime analysis, we consider the (1+1)-ENAS algorithm based on (1+1)-EA with mutation only, which serves as a theoretical foundation for runtime analysis and algorithm design. Its main steps are:

1. Randomly sample $n_z^m$ ($m \in [1..M-1], z \in \{A, B, C\}$) in the initial solution according to the uniform distribution $U[1, s]$, where $s \in \mathbb{N}^+$ is an algorithm-specific parameter.

2. Generate offspring by executing mutation on a parent.

3. Select the solution with higher fitness to enter into the next iteration. In case of ties, the offspring is preferred to the parent by the selection operator.

4. Repeat Steps 2 and 3 until an optimal solution is found.

In the above steps, mutation plays a key role in guiding the search process, with two commonly used types in the ENAS community: the one-bit mutation that changes only one position, and the bit-wise mutation that changes each position with a certain probability. These two mutations typically affect the runtime of the algorithm, which will be separately examined in the (1+1)-ENAS algorithm. Specifically, we employ a "two-level" mutation process: an outer-level mutation that selects cells for mutation and an inner-level mutation that alters blocks within the selected cells. The outer-level mutation can be either one-bit mutation, which mutates one randomly selected cell, or bit-wise mutation, which mutates each cell independently with probability $1/(M-1)$. The inner-level mutation, executed after the outer-level mutation, can be either *local* (mutating each selected cell once) or *global* (mutating $K$ times, with $K \sim \text{Pois}(1)$) (Kratsch et al., 2010; Durrett et al., 2011; Qian et al., 2015; 2023) mutation. For each time of mutation, the algorithm applies the operation defined in Definition 2.1.

**Definition 2.1** (Mutation Operation). Given any cell $(n_A^m, n_B^m, n_C^m)$ of a solution, the algorithm randomly selects a block type $V \in \{A, B, C\}$ and applies one of the following operations uniformly at random: (1) **Addition:** adds a $V$-type block (i.e., $n_V^m \leftarrow n_V^m + 1$); (2) **Deletion:** deletes a $V$-type block (i.e., $n_V^m \leftarrow n_V^m - 1$) when

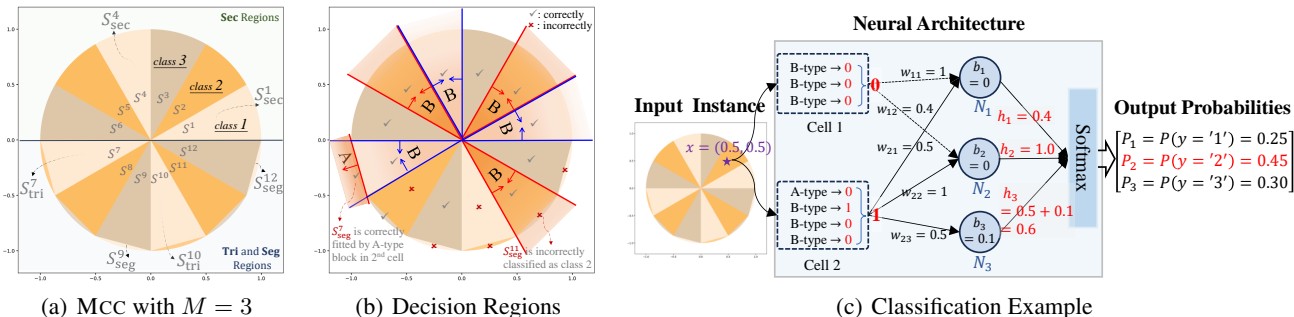

|  (a) MCC with $M = 3$  |  (b) Decision Regions  |  (c) Classification Example  |

*Figure 2.* Illustration of neural architecture solving MCC. (a) MCC with $M = 3$ and $r = 2$, depicting a 3-class classification problem. The points labeled as class 1, 2, and 3 are colored off-white, yellow, and khaki, respectively. Each class (e.g., class 1) has two segments (e.g., $S_{\text{seg}}^9 \cup S_{\text{seg}}^{12}$), two sectors (e.g., $S_{\text{sec}}^1 \cup S_{\text{sec}}^4$), and two triangles (e.g., $S_{\text{tri}}^7 \cup S_{\text{tri}}^{10}$). (b) Decision regions produced by neural architecture $\{(0, 3, 0), (1, 3, 0)\}$: blue arrows point to class 1 (excluding the red-arrow regions), red arrows point to class 2, and the remaining regions to class 3. The regions marked with a "✓" are correctly classified. (c) Classification example for input instance $(0.5, 0.5) \in S_{\text{Sec}}^2$ using neural architecture $\{(0, 3, 0), (1, 3, 0)\}$. The first cell outputs 0 since it has no block to cover the region $S_{\text{Sec}}^2$, while the second cell outputs 1 since it has a B-type block to cover the region $S_{\text{Sec}}^2$. The neurons $\{N_1, N_2, N_3\}$ then output values of 0.4, 1.0, and 0.6, respectively. The softmax layer converts these into probabilities of 0.25, 0.45, 0.3, and class 2 (0.45) is chosen as the classification result.

$n_V^m > 0$; (3) **Modification:** randomly selects a block type $W \in \{A, B, C\} \backslash V$ and then performs Deletion on $V$ while performs Addition on $W$ when $n_V^m > 0$ (i.e., $n_V^m \rightarrow n_V^m - 1$ and $n_W^m \rightarrow n_W^m + 1$); otherwise, $n_V^m$ and $n_W^m$ remain unchanged.

Notably, there is a sequence in the two-level mutation process, i.e., first selecting the cell to mutate (outer-level mutation) and then applying the mutation (inner-level mutation) to the selected cell. Reversing the sequence would make the algorithm infeasible, as the index of the cell to be mutated would be unknown, making it unclear where the inner-level mutation should be applied. Specifically, given a solution $\boldsymbol{x}$, which is encoded by $M - 1$ triplets of integers, i.e., $\boldsymbol{x} = \{(n_A^1, n_B^1, n_C^1), \ldots, (n_A^{M-1}, n_B^{M-1}, n_C^{M-1})\}$, the algorithm must first select the cell index $m \in \{1, \ldots, M-1\}$, and then apply the inner-level mutation to modify the selected cell $(n_A^m, n_B^m, n_C^m)$.

# 3. A Multiclass Classification Problem

The differences of the classification problem and the binary classification problem (Lv et al., 2024b) for ENAS algorithms include the following: 1) decision regions: multiclass classification divides input space into $M$ decision regions (vs. two in binary classification), increasing neural architectural demands; 2) classification accuracy: multiclass classification aggregates per-class accuracy across all $M$ regions, amplifying the complexity of the fitness evaluation; 3) search space: the neural architecture for solving multiclass classification is a combination of multiple binary classifiers or a more complex architecture (vs. binary classification's binary classifier), exponentially expanding the search space.

These differences introduce three challenges: 1) problem definition: accurately modeling inter-class dependencies and region-specific sample distributions; 2) fitness function: mathematically formulating the fitness (i.e., classification accuracy) of neural architectures from geometric properties; 3) search space partition: partitioning the search space by analyzing the interactions between architectural components (e.g., blocks, cells).

To tackle the above challenges, this section defines a multiclass classification problem named MCC and a fitness function that evaluates the quality of neural architectures solving it. The search space of the problem is then partitioned to facilitate runtime analysis.

## 3.1. Problem Definition

We propose a multiclass classification benchmark problem MCC that captures essential properties of real-world multiclass classification tasks, including both linearly and nonlinearly divisible decision regions (e.g., half-space region, unbounded/bounded polyhedra region). As defined in Definition 3.1, the input is a 2-dimensional input vector as suggested in the binary classification benchmark problems (Fischer et al., 2023; Lv et al., 2024b); the output label (class) takes on one of $M$ labels.

**Definition 3.1** (MCC Problem). Let the inputs consist of the points on the unit circle $S := \{x \in \mathbb{R}^2 \mid \|x\|_2 \leq 1\}$, with each point labeled as class $y \in [1..M]$, where $M \geq 2$. The unit circle $S$ is evenly divided into $n = 2rM$ sectors with angle $\frac{2\pi}{n}$, where $r \geq 2$. Thus, $S = \cup_{k=1}^n S_{\text{sec}}^k$, with $S_{\text{sec}}^k$ representing the $k$-th sector. Each sector $S_{\text{sec}}^k$ can be further divided into a triangle $S_{\text{tri}}^k$ with area $\frac{1}{2} \sin(\frac{2\pi}{n})$ and

a segment (a segment is cut from a sector by a chord) $S_{\text{seg}}^k$ with area $\frac{\pi}{n} - \frac{1}{2}\sin(\frac{2\pi}{n})$. Let the points with label $m$ be distributed across $r$ segmenets, $r$ sectors, and $r$ triangles, with the corresponding regions denoted as $\text{Seg}^m$, $\text{Sec}^m$, and $\text{Tri}^m$, respectively. These regions are defined as follows:

$$\text{Sec}^m = \bigcup_{k=1}^{r} S_{\text{sec}}^{m+(k-1)M}, \quad \text{Tri}^m = \bigcup_{k=r+1}^{2r} S_{\text{tri}}^{m+(k-1)M},$$

$$\text{Seg}^{m\neq 1} = \bigcup_{k=r+1}^{2r} S_{\text{seg}}^{(m-1)+(k-1)M}, \text{Seg}^1 = \bigcup_{k=r+1}^{2r} S_{\text{seg}}^{kM}.$$

Then, the set of points with label $m \in [1..M]$ is

$$\mathcal{L}^m = \text{Seg}^m \cup \text{Sec}^m \cup \text{Tri}^m.$$

An example of MCC ($M = 3$, $r = 2$, $n = 12$) is shown in Figure 2(a). The problem has the following characteristics: (1) The decision region of a class is the disjoint union of $r$ half-spaces (segments), $r$ unbounded polyhedra (sectors), and $r$ bounded polyhedra (triangles). (2) Triangles and sectors of the same class are never connected. For example, in Figure 2(a), triangle $S_{\text{tri}}^7$ of class 1 is bordered by a sector of class 2 and a triangle of class 3. (3) Segment $S_{\text{seg}}^i$ connected to triangle $S_{\text{tri}}^i$ of class $m$ must belong to class $(m + 1) \mod M$. For example, $S_{\text{tri}}^7$ in Figure 2(a) belongs to class 1, whereas $S_{\text{seg}}^7$ belongs to class 2. In our work, we explore how fast does the ENAS algorithm can find an optimal neural architecture that correctly forms all of the decision regions in the MCC problem.

We consider the neural architecture described in Section 2.1 to solve the MCC problem. Figure 2(b) depicts the decision regions of the neural architecture $\{(n_A^1 = 0, n_B^1 = 3, n_C^1 = 0), (n_A^2 = 1, n_B^2 = 3, n_C^2 = 0)\}$, which visually demonstrates how neural architecture solves the problem shown in Figure 2(a). The decision region for class 1 excludes $S_{\text{seg}}^7$ due to the use of an A-type block in the second cell, which leads to this region being classified as class 2. This highlights how the neural architecture addresses the complex inter-class relationship in multi-class classification. Based on the above, Figure 2(c) presents an example of how the neural architecture classifies a specific input instance, with detailed computation from input to output. Additional classification examples can be found in **Appendix B**.

The classification accuracy $\text{acc}(\mathbf{H})$ of a parameterized neural architecture $\mathbf{H}$ can be defined as the ratio of correctly classified points by $\mathbf{H}$ to the total number of points in the unit circle $S$, which is intuitive, easy to understand, and suitable for balanced datasets because it does not consider the class distribution (Grandini et al., 2020). Thus, we have

$$\text{acc}(\mathbf{H}) = \frac{\text{vol}\left(\left(\bigcup_{m=1}^{M-1}(\mathcal{C}^m \cap \mathcal{L}^m)\right) \cup \left(\bigcap_{m=1}^{M-1}\overline{\mathcal{C}^m} \cap \mathcal{L}^M\right)\right)}{\text{vol}(S)}, \quad (1)$$

where $\mathcal{C}^m$ is the set of points that are covered by the blocks in the $m$-th cell of the neural architecture, $\mathcal{L}^m$ represents the set of points with label $m$, $\overline{\mathcal{C}^m} = \mathbb{R}^2 \backslash \mathcal{C}^m$, and $\text{vol}(\cdot)$ denotes the area. $\bigcup_{m=1}^{M-1}(\mathcal{C}^m \cap \mathcal{L}^m)$ represents the set of points that are correctly classified in classes 1 to $M - 1$, and $\left(\bigcap_{m=1}^{M-1}\overline{\mathcal{C}^m} \cap \mathcal{L}^M\right)$ is the set of points that are correctly classified in class $M$. For example, in the case of Figure 2(b), we have $\left(\bigcap_{m=1}^{M-1}\overline{\mathcal{C}^m} \cap \mathcal{L}^M\right) = \mathcal{L}^M \backslash S_{\text{seg}}^{11}$, as the second cell misclassifies $S_{\text{seg}}^{11}$ of class 3; the classification accuracy can be calculated according to Eq. (1), i.e., $((4 \cdot \text{Ar}_{\text{sec}} + 2 \cdot \text{Ar}_{\text{tri}} + 1 \cdot \text{Ar}_{\text{seg}}) + (2 \cdot \text{Ar}_{\text{sec}} + 2 \cdot \text{Ar}_{\text{tri}} + 1 \cdot \text{Ar}_{\text{seg}}))/\pi \approx 0.83$.

In general, three kinds of regions need to be fit in each class: (1) $a = r$ segment regions (A-type), (2) $b = r$ sector regions (B-type, or A-type plus C-type), and (3) $c = r$ triangle regions (C-type). From the perspective of neural architecture optimization, there are six important block counts to consider in each cell of neural architecture:

1) $n_A'$: Number of A-type blocks that classify segments.

2) $n_A''$: Number of A-type blocks that classify sectors (together with a C-block).

3) $n_B'$: Number of B-type blocks that classify sectors alone.

4) $n_C'$ : Number of C-type blocks that classify sectors (together with an A-type block).

5) $n_C''$ : Number of C-type blocks that classify triangles.

6) $n_B''$: Number of B-type blocks that correctly classify the triangles but incorrectly classify the segments.

For the $(M - 1)$-th cell, when $n_B < b$, the missing $b - n_B$ B-type blocks need to be compensated by A-type and C-type blocks. In this case, we need $n_A'' \geq b - n_B$ (i.e., $n_A \geq a + b - n_B$) and $n_C' \geq b - n_B$ (i.e., $n_C \geq c + b - n_B$). In addition, if $n_B > b$ and $n_C < c$, then there must be $n_B'' > 0$ segments in $\text{Seg}^M$ that are misclassified as class $M - 1$, due to insufficient C-type blocks. To avoid this, we must ensure $n_B'' = 0$, i.e., we need $n_C \geq n_C'' \geq c$ to ensure that all triangles of class $M - 1$ are correctly classified by C-type blocks. Overall, the $(M - 1)$-th cell of the optimum needs to satisfy the following conditions:

- $n_A \geq a + \max\{0, b - n_B\} = \max\{r, 2r - n_B\}$,

- $n_B + n_C \geq b + c = 2r$,

- $n_C \geq c = r$.

These conditions must also be satisfied for the other cells; otherwise, some points will be misclassified as class $M$.

## 3.2. Mathematically Formulated Fitness Function

ENAS primarily focuses on the evolution of neural architecture, leaving the fitness evaluation of architecture to learning algorithms (such as the gradient-based algorithm) (Elsken et al., 2019). However, the practical optimization methods may not guarantee the achievement of the optimal parameters. To analyze ENAS purely, we assume that the optimal or approximate parameters of each evolved architecture can be achieved in fitness evaluation, as we discussed in Section 2.1. Based on this assumption, we can mathematically formulate a fitness function to describe how the basic units (i.e., cells and blocks) of neural architecture influence its fitness on the MCC problem.

Before formulating the fitness function, we define two key quantities related to the solution $\boldsymbol{x}$, which reflect the number of decision regions that the solution can form. One is the number of decision regions shaped as triangles that can be correctly formed by the $m$-th cell of $\boldsymbol{x}$, which is denoted as $I_{\boldsymbol{x}}^m$ and can be calculated by

$$I_{\boldsymbol{x}}^m = \min\{n_B^m + n_C^m, b + c\} \in [0..2r]. \qquad (2)$$

The other is the number of decision regions shaped as segments that can be correctly formed by the $m$-th cell of $\boldsymbol{x}$, which is denoted as $J_{\boldsymbol{x}}^m$ and can be calculated by

$$J_{\boldsymbol{x}}^m = \min\{n_B^m, b\} + \min\{n_A^m, a + \max\{b - n_B^m, 0\}\}, \qquad (3)$$

where the first term represents up to $b$ B-type blocks for classifying the segments within the sectors $\mathrm{Sec}^m$, and the second term indicates that at most $a + \max\{b - n_B^m, 0\}$ A-type blocks are used to classify the segments in $\mathrm{Seg}^m$ and the segments within the sectors $\mathrm{Sec}^m$. Consequently, $J_{\boldsymbol{x}}^m \in [0..2r]$.

Next, we establish the relationship between the decision regions and the classification accuracy of a solution, based on the classification accuracy calculation shown in Eq. (1). The first term in the numerator of Eq. (1) reflects how many points belonging to class $[1..M-1]$ are correctly classified by $\boldsymbol{x}$. This is related to the number of correctly classified triangles and segments (including those within sectors) that belong to classes $[1..M-1]$, respectively, denoted as

$$\mathbb{I}_{\boldsymbol{x}} = \sum_{m=1}^{M-1} I_{\boldsymbol{x}}^m \in [0..n - 2r],$$

$$\mathbb{J}_{\boldsymbol{x}} = \sum_{m=1}^{M-1} J_{\boldsymbol{x}}^m \in [0..n - 2r]. \qquad (4)$$

The second term in the numerator of Eq. (1) reflects how many points belonging to class $M$ can be classified correctly by $\boldsymbol{x}$. Any point in $\mathrm{Sec}^M \cup \mathrm{Tri}^M$ can always be correctly classified. This is because no blocks in $\boldsymbol{x}$ cover these points,

resulting in all cells outputting 0-bits, which allows only the neuron $N_M$ with a non-zero bias to output a value greater than 0 (specifically, 0.1). However, point in $\mathrm{Seg}^M$ may be misclassified by the $(M-1)$-th cell. Specifically, if the value $n_B''$ of the $(M-1)$-th cell is greater than 0, then $\min\{(n_B^{M-1} - b), (c - n_C^{M-1})\}$ segments of class $M$ will be incorrectly classified; otherwise, all segments of class $M$ are correctly classified. Let $\epsilon_{\boldsymbol{x}}$ denote the number of misclassified segments of class $M$. Then, we have

$$\epsilon_{\boldsymbol{x}} = \max\{0, \min\{(n_B^{M-1} - b), (c - n_C^{M-1})\}\} \in [0..r]. \qquad (5)$$

Consequently, $b + c = 2r$ triangles and $a + b - \epsilon_{\boldsymbol{x}} = 2r - \epsilon_{\boldsymbol{x}}$ segments of class $M-1$ can be correctly classified.

By combining the two terms in the numerator of Eq. (1), we can derive the number of triangles and segments (including those within sectors) that can be correctly classified by $\boldsymbol{x}$. Specifically, $\mathbb{I}_{\boldsymbol{x}} + 2r$ triangles and $\mathbb{J}_{\boldsymbol{x}} + 2r - \epsilon_{\boldsymbol{x}}$ segments are correctly classified. Additionally, we have $\mathbb{J}_{\boldsymbol{x}} + 2r - \epsilon_{\boldsymbol{x}} \in [2r, n]$, where the lower bound of $2r$ holds because $\mathbb{J}_{\boldsymbol{x}} - \epsilon_{\boldsymbol{x}} \geq J_{\boldsymbol{x}}^{M-1} - \epsilon_{\boldsymbol{x}} \geq 0$. Then, we introduce Lemma 3.2 to evaluate the fitness of any given neural architecture.

**Lemma 3.2** (Fitness Function). *Given a neural architecture $\boldsymbol{x}$, its fitness value for solving* MCC *can be calculated by*

$$\mathscr{F}(\boldsymbol{x}) = (\mathrm{Ar}_{\mathrm{tri}} \cdot (\mathbb{I}_{\boldsymbol{x}} + 2r) + \mathrm{Ar}_{\mathrm{seg}} \cdot (\mathbb{J}_{\boldsymbol{x}} + 2r - \epsilon_{\boldsymbol{x}})) / \pi, \qquad (6)$$

*where $\mathrm{Ar}_{\mathrm{tri}}$ and $\mathrm{Ar}_{\mathrm{seg}}$ denote the area of a triangle and a segment, respectively, and $\pi$ is the area of the unit circle.*

*Proof of Lemma 3.2.* Let $\mathscr{F}(\boldsymbol{x})$ denote the fitness value of architecture $\boldsymbol{x}$. According to the formulation for the calculation of classification accuracy (i.e., Eq. (1) in the main paper), we have

$$\mathscr{F}(\boldsymbol{x}) = \left(\sum_{m=1}^{M-1} f_m + (\pi/M - \mathrm{Ar}_\epsilon)\right) / \pi, \qquad (7)$$

where $f_m$ represents the area of regions that are correctly classified by the $m$-th cell of $\boldsymbol{x}$, $\pi$ is the area of the unit circle, and $\pi/M - \mathrm{Ar}_\epsilon$ denotes the area of regions belonging to class $M$ minus the area of regions that are misclassified by the $(M-1)$-th cell.

According to the parameter settings (as discussed in Section 2.1 of the main paper), the area of the regions that can be correctly classified by an A-type, B-type, and C-type block is $\{0, \mathrm{Ar}_{\mathrm{seg}}\}$, $\{0, \mathrm{Ar}_{\mathrm{tri}}, \mathrm{Ar}_{\mathrm{sec}}\}$, and $\{0, \mathrm{Ar}_{\mathrm{tri}}\}$, respectively. We note that a sector can be divided into a triangle and a segment, as $\mathrm{Ar}_{\mathrm{sec}} = \mathrm{Ar}_{\mathrm{seg}} + \mathrm{Ar}_{\mathrm{tri}}$. Since $I_{\boldsymbol{x}}^m$ and $J_{\boldsymbol{x}}^m$ represent the number of correctly covered triangles and segments, respectively, $\forall m \in [1..M-1]$, we have

$$f_m = I_{\boldsymbol{x}}^m \cdot \mathrm{Ar}_{\mathrm{tri}} + J_{\boldsymbol{x}}^m \cdot \mathrm{Ar}_{\mathrm{seg}}.$$

In addition, there are $\epsilon_{\boldsymbol{x}}$ segments in class $M$ that can be misclassified by the $(M-1)$-th cell, so we have $\mathrm{Ar}_\epsilon =$

$\epsilon_{\boldsymbol{x}} \cdot \mathrm{Ar_{seg}}$. Therefore, Eq. (7) can be expressed as

$$\mathscr{F}(x) = \sum_{m=1}^{M-1} \frac{I_{\boldsymbol{x}}^m \cdot \mathrm{Ar_{tri}} + J_{\boldsymbol{x}}^m \cdot \mathrm{Ar_{seg}}}{\pi} + \frac{1}{M} - \frac{\mathrm{Ar_\epsilon}}{\pi}$$

$$= \mathbb{I}_x \cdot \frac{\mathrm{Ar_{tri}}}{\pi} + \mathbb{J}_x \cdot \frac{\mathrm{Ar_{seg}}}{\pi} + \frac{1}{M} - \epsilon_{\boldsymbol{x}} \cdot \frac{\mathrm{Ar_{seg}}}{\pi}$$

$$= \frac{1}{M} + (\mathbb{I}_x \cdot \mathrm{Ar_{tri}} + (\mathbb{J}_x - \epsilon_{\boldsymbol{x}}) \cdot \mathrm{Ar_{seg}}) \frac{1}{\pi},$$

where $\mathbb{I}_x = \sum_{m=1}^{M-1} I_{\boldsymbol{x}}^m$ and $\mathbb{J}_x = \sum_{m=1}^{M-1} J_{\boldsymbol{x}}^m$. Since $n = 2Mr$ and $\mathrm{Ar_{tri}} + \mathrm{Ar_{seg}} = \pi/n$, the lemma holds. $\qquad\square$

To illustrate how the fitness function defined in Eq. (6) evaluates a solution, we provide a calculation example by using the neural architecture shown in Figure 2(b) as $\boldsymbol{x}$. In this case, we have $\epsilon_{\boldsymbol{x}} = 1$, $\mathbb{I}_{\boldsymbol{x}} = 6$, $\mathbb{J}_{\boldsymbol{x}} = 6$, and $\mathrm{Ar_{sec}} = \mathrm{Ar_{tri}} + \mathrm{Ar_{seg}} = \pi/n$. By substituting these into Eq. (6), we can yield the same accuracy of 0.83.

The mathematically formulated fitness function in Eq. (6) demonstrates that the fitness of $\boldsymbol{x}$ can be improved by appropriately increasing the number of A-type, B-type, and C-type blocks in each cell, as these contribute positively to both $\mathbb{I}_{\boldsymbol{x}} + 2r$ and $\mathbb{J}_{\boldsymbol{x}} + 2r - \epsilon_{\boldsymbol{x}}$, which correspond to the number of correctly classified triangles and segments (including those within sectors) for classes 1 to $M$, respectively. This behavior is somewhat similar to the ONEMAX benchmark function for EAs, allowing it to play the role of ONEMAX in ENAS.

### 3.3. Search Space Partition

In the runtime analysis of EAs, it is common to construct a distance function to investigate the progress of the algorithm (Auger & Doerr, 2011; Neumann & Witt, 2010; Zhou et al., 2019; Doerr & Neumann, 2020). The partitioning of the search space plays a key role in defining the distance function. The runtime analysis of ENAS follows the conventional analysis approach. Therefore, we present a partition of the search space designed in Section 2.2, which can facilitate the runtime analysis of ENAS solving MCC.

Let $\mathcal{S}$ denote the search space, which also refers to the solution space. According to the first term in Eq. (6), i.e., $\mathbb{I}_{\boldsymbol{x}} + 2r \in [2r..n]$, $\mathcal{S}$ can be partitioned into $n - 2r + 1$ subspaces that are denoted as $\cup_{i=0}^{n-2r} \mathcal{S}_i$, where $\mathcal{S}_i = \{\boldsymbol{x} \in \mathcal{S} \mid \mathbb{I}_{\boldsymbol{x}} + 2r = i + 2r\}$. The following relationship exists between the subspaces:

$$\mathcal{S}_0 <_{\mathscr{F}} \mathcal{S}_1 <_{\mathscr{F}} \mathcal{S}_2 < \cdots <_{\mathscr{F}} \mathcal{S}_{n-2r},$$

where $\mathcal{S}_i <_{\mathscr{F}} \mathcal{S}_{i+1}$ represents that $\mathscr{F}(\boldsymbol{x}) < \mathscr{F}(\boldsymbol{y})$ for all $\boldsymbol{x} \in \mathcal{S}_i$ and all $\boldsymbol{y} \in \mathcal{S}_{i+1}$. The basis for the partition is that the fitness function in Eq. (6) is dominated by the $\mathbb{I}_{\boldsymbol{x}}$-term, as $\mathrm{Ar_{tri}} > (n - 2r) \cdot \mathrm{Ar_{seg}}$, where $n \geq 8$ in the MCC problem.

In addition, according to the second term in Eq. (6), i.e., $\mathbb{J}_{\boldsymbol{x}} + 2r - \epsilon \in [2r..n]$, $\mathcal{S}_i$ can be partitioned into $(n - 2r + 1)$ sub-subspaces that are denoted as $\cup_{j=0}^{n-2r} \mathcal{S}_i^j$, where $\mathcal{S}_i^j = \{\boldsymbol{x} \in \mathcal{S} \mid \mathbb{I}_{\boldsymbol{x}} = i, \mathbb{J}_{\boldsymbol{x}} + 2r - \epsilon_{\boldsymbol{x}} = j + 2r\}$. The following relationship exists between different sub-subspaces of $\mathcal{S}_i$:

$$\mathcal{S}_i^0 <_{\mathscr{F}} \mathcal{S}_i^1 <_{\mathscr{F}} \mathcal{S}_i^2 < \cdots <_{\mathscr{F}} \mathcal{S}_i^{n-2r},$$

where $\mathcal{S}_i^j <_{\mathscr{F}} \mathcal{S}_i^{j+1}$ represents that $\mathscr{F}(\boldsymbol{x}) < \mathscr{F}(\boldsymbol{y})$ for all $\boldsymbol{x} \in \mathcal{S}_i^j$ and all $\boldsymbol{y} \in \mathcal{S}_i^{j+1}$.

## 4. Runtime Analysis

In this section, we analyze the expected runtime of (1+1)-ENAS for solving MCC. As the (1+1)-ENAS algorithm generates only one offspring solution in each generation, its expected runtime equals the expected number $T$ of generations to reach the optimum, which is denoted as $\mathbb{E}[T]$.

### 4.1. One-bit Mutation

We prove in Theorems 4.1 and 4.2 that the upper and lower bounds on the expected runtime of (1+1)-ENAS$_\mathrm{onebit}$ solving MCC are $\mathbb{E}[T] = O\left(rM \ln(rM)\right)$ and $\mathbb{E}[T] = \Omega(rM \ln M)$, respectively. Note that (1+1)-ENAS$_\mathrm{onebit}$ refers to (1+1)-ENAS using one-bit outer-level mutation and either local or global inner-level mutation.

**Theorem 4.1.** *(Upper bound) The (1+1)-ENAS$_\mathrm{onebit}$ algorithm needs $\mathbb{E}[T] = O\left(rM \ln(rM)\right)$ to find the optimum for* MCC.

**Theorem 4.2.** *(Lower bound) When the upper bound s on the number of each type of block in the initial solution is $r$, the (1+1)-ENAS$_\mathrm{onebit}$ algorithm needs $\mathbb{E}[T] = \Omega(rM \ln M)$ to find the optimum for* MCC.

Before proving Theorem 4.1, we briefly outline the main idea of the proof. We divide the optimization procedure into two phases, where the first aims at finding one solution that belongs to $\mathcal{S}_{n-2r}$, and the second aims at finding one optimal solution that belongs to $\mathcal{S}_{n-2r}^{n-2r}$. For each phase, we utilize the multiplicative drift analysis (Doerr et al., 2012) to derive its expected runtime. This widely used mathematical tool is well-suited for cases where the expected progress is proportional to the current objective value, as observed during the proof process. To utilize this tool, we construct a distance function $V(\cdot)$ for each phase to measure how far a solution is from the target solution of the phase. For the simplicity of presentation, we let $n - 2r = 2r(M - 1)$ be expressed as $N$, $\gamma_{i,i'}$ denote the probability that $I_{\boldsymbol{x}}^m = i$ changes to $I_{\boldsymbol{x}}^m = i'$ after the inner-level mutation, and $\eta_{z,z'}$ be the probability that $\mathbb{J}_{\boldsymbol{x}} - \epsilon_{\boldsymbol{x}}$ increases by $z' - z$ after the inner-level mutation.

In addition, to identify which cells contribute to progress toward the optimum, we utilize a bit-string of length $M - 1$

to characterize whether each cell's $I_{\boldsymbol{x}}^m$ (or $J_{\boldsymbol{x}}^m$) matches the optimum or not, denoted as $\boldsymbol{o} = \{o^1, \ldots, o^{M-1}\} \in \{0,1\}^{M-1}$, where $o^m$ ($m < M-1$) can be expressed by

$$o^m = \begin{cases} 1 & \text{if } (\mathbb{I}_{\boldsymbol{x}} < N, I_{\boldsymbol{x}}^m = 2r) \text{ or } (\mathbb{I}_{\boldsymbol{x}} = N, J_{\boldsymbol{x}}^m = 2r), \\ 0 & \text{if } (\mathbb{I}_{\boldsymbol{x}} < N, I_{\boldsymbol{x}}^m < 2r) \text{ or } (\mathbb{I}_{\boldsymbol{x}} = N, J_{\boldsymbol{x}}^m < 2r), \end{cases}$$

and $o^{M-1} = 1$ when $(\mathbb{I}_{\boldsymbol{x}} < N, I_{\boldsymbol{x}}^{M-1} = r)$ or $(\mathbb{I}_{\boldsymbol{x}} = N, J_{\boldsymbol{x}}^{M-1} = 2r, \epsilon_{\boldsymbol{x}} = 0)$; otherwise, $o^{M-1} = 0$. The number of cells that match the optimum is given by the number of 1-bits in the binary string $\boldsymbol{o}$, denoted as $|\boldsymbol{o}|_1 = \sum_{m=1}^{M-1} o^m$. Then, $|\boldsymbol{o}|_0$ denotes the number of 0-bits. Given any neural architecture with $\mathbb{I}_{\boldsymbol{x}} = i$, we have the bounds of $|\boldsymbol{o}|_1$, i.e.,

$$\max\{i - (M-1)(2r-1), 0\} \leq |\boldsymbol{o}|_1 \leq \frac{i}{2r}, \quad (8)$$

where the upper bound holds by the maximum value of $I_{\boldsymbol{x}}^m$ being $2r$, and the lower bound is occured when $\forall m: I_{\boldsymbol{x}}^m \geq 2r-1$. Given $\mathbb{J}_{\boldsymbol{x}} = j$ and $\mathbb{J}_{\boldsymbol{x}} - \epsilon_{\boldsymbol{x}} = z$, we have $|\boldsymbol{o}|_1 \leq \lfloor (j - J_{\boldsymbol{x}}^{M-1})/(2r) \rfloor + \lfloor (J_{\boldsymbol{x}}^{M-1} - (j-z))/(2r) \rfloor \leq z/(2r)$. The bounds will be utilized in the proofs of Theorems 4.1– 4.4. Next, we proceed to establish the proof for Theorem 4.1.

*Proof of Theorem 4.1.* According to the partition of search space, we divide the optimization procedure into two phases, and we pessimistically assume that the solution with $\mathbb{J}_{\boldsymbol{x}} > 0$ is not found in the first phase.

- Phase 1: This phase starts after initialization and ends when a solution $\boldsymbol{x}$ with $\mathbb{I}_{\boldsymbol{x}} = n - 2r$ is found, i.e., $\boldsymbol{x} \in \mathcal{S}_{n-2r}$.

- Phase 2: This phase starts after Phase 1 and ends when a solution $\boldsymbol{x} \in \mathcal{S}_{n-2r}^{n-2r}$ is found.

**Phase 1.** We define the distance function $V_1(\boldsymbol{x}) = N - \mathbb{I}_{\boldsymbol{x}} = \sum_{m=1}^{M-1}(2r - I_x^m)$ to measure the progress towards the optimum in the first phase. It is easy to verify that $V_1(\boldsymbol{x}) = 0$ iff $\mathbb{I}_{\boldsymbol{x}} = N$. To make progress, it is sufficient to select one cell with $o^m = 0$ and make the value $I_{\boldsymbol{x}}^m$ increase by executing the inner-level mutation (local or global). The probability of the former event (selecting one cell with $o^m = 0$) equals $|\boldsymbol{o}|_0/(M-1)$ since there are $|\boldsymbol{o}|_0$ cells with $o^m = 0$. Hence, the probability of making progress is at least $(|\boldsymbol{o}|_0/(M-1)) \cdot \gamma_{i,i+1}$. In this case, the distance $V_1$ can decrease by at least 1.

Then, we analyze the expectation of the one-step progress (i.e., drift) towards the target solution in the first phase. We have

$$\mathbb{E}[V_1(\boldsymbol{x}_t) - V_1(\boldsymbol{x}_{t+1}) \mid V_1(\boldsymbol{x}_t)]$$
$$\geq \frac{|\boldsymbol{o}|_0}{M-1} \cdot \gamma_{i,i+1} \cdot 1 \geq \left(1 - \frac{\mathbb{I}_{\boldsymbol{x}_t}}{N}\right) \cdot \frac{2}{9}, \quad (9)$$

where the second inequality holds by $|\boldsymbol{o}|_0 = M-1-|\boldsymbol{o}|_1 \geq M - 1 - \mathbb{I}_{\boldsymbol{x}_t}/(2r)$ (derived by Eq. (8)) and $\gamma_{i,i+1} \geq 2/9$ (Lv *et al.* (2024b) showed that: if the inner-level mutation adopts the local mutation, $\gamma_{i,i+1} \geq 2/9$ as shown in their Lemma 3; if the inner-level mutation adopts the global mutation, $\gamma_{i,i+1} \geq 0.23$ as shown in their Lemma 5). In other words, the drift of $V_1(\boldsymbol{x}_t)$ is bounded from below by $\sigma \cdot V_1(\boldsymbol{x}_t)$ with $\sigma = 2/(9N)$. Note that $V_1(x_t) = N - \mathbb{I}_{\boldsymbol{x}_t}$.

Owing to $0 < \mathbb{I}_{\boldsymbol{x}} < N$ during the first phase, the distance of the initial individual is $V_1(\boldsymbol{x}_0) \in [1..N-1]$. By the multiplicative drift theorem (Doerr et al., 2012), the expected runtime of finding a solution with $V_1 = 0$ (i.e., $\mathbb{I}_{\boldsymbol{x}} = N$) is $O((1 + \ln V_1(\boldsymbol{x}_0))/\sigma) = O(N \ln N)$.

**Phase 2.** After the aforementioned phase, we pessimistically assume that the searched solution with $\mathbb{J}_{\boldsymbol{x}} = 0$, which means that $|\boldsymbol{o}|_1 = 0$. As the goal of the second phase is to reach $\mathbb{J}_{\boldsymbol{x}} - \epsilon_{\boldsymbol{x}} = N$, we construct a new distance function $V_2(\boldsymbol{x}) = N - (\mathbb{J}_{\boldsymbol{x}} - \epsilon_{\boldsymbol{x}})$. It is easy to verify that $V_2(\boldsymbol{x}) = 0$ if and only $\mathbb{J}_{\boldsymbol{x}} - \epsilon_{\boldsymbol{x}} = N$.

The analysis is similar to phase 1. One difference is the constant probability $\eta_{z,z'}$, which is lower bounded by $1/9$ as shown in Lemmas 4 and 5 of (Lv et al., 2024a). Similar to the calculation in Eq. (9), the expected one-step progress in $V_2(\boldsymbol{x}_t)$ is at least $(V_x(\boldsymbol{x}_t)/N) \cdot (1/9)$ since $|\boldsymbol{o}|_0 = M - 1 - |\boldsymbol{o}|_1 \geq M - 1 - (\mathbb{J}_{\boldsymbol{x}_t} - \epsilon_{\boldsymbol{x}_t})/(2r)$. By utilizing the multiplicative drift, the expected runtime for the second phase is $O(N \ln N)$. By combining the two phases, we get $\mathbb{E}[T] \in O\left(rM \ln(rM)\right)$ because $N = 2r(M-1)$. $\square$

The proof idea of Theorem 4.2 is that the initial solution has $(M-1)/r$ 0-bits in $\boldsymbol{o}$ with probability at least $1/2$, and then (1+1)-ENAS$_{\text{onebit}}$ requires at least $\Omega(rM \ln M)$ expected runtime to find the solution belonging to $\mathcal{S}_{n-2r}$. This is because the probability that at least one of these 0-bits in $\boldsymbol{o}$ of the initial solution remains unchanged over $t = \max\{1, M-2\} \cdot \ln(M-1) \cdot r$ generations is lower bounded by $1 - e^{-1}$. The complete proof of Theorem 4.2 is provided in **Appendix C.1**.

### 4.2. Bit-wise Mutation

Next, we will show that by replacing one-bit with bit-wise mutation, the (1+1)-ENAS algorithm can achieve a similar runtime. In particular, we prove in Theorems 4.3 and 4.4 that the upper and lower bounds on the expected runtime of (1+1)-ENAS$_{\text{bitwise}}$ solving MCC are $\mathbb{E}[T] = O(rM \ln(rM))$ and $\mathbb{E}[T] = \Omega(rM \ln M)$, respectively. Note that (1+1)-ENAS$_{\text{bitwise}}$ denotes that (1+1)-ENAS uses bit-wise outer-level mutation and either local or global inner-level mutation.

**Theorem 4.3.** *(Upper bound) The (1+1)-ENAS$_{\text{bitwise}}$ algorithm needs $\mathbb{E}[T] = O(rM \ln(rM))$ to find the optimum*

*for* MCC.

**Theorem 4.4.** *(Lower bound) When the upper bound s on the number of each type of block in the initial solution is r, the (1+1)-ENAS*$_{\text{bitwise}}$ *algorithm needs* $\mathbb{E}[T] = \Omega(rM \ln M)$ *to find the optimum for* MCC.

The main proof idea of Theorem 4.3 is similar to that of Theorem 4.1, i.e., partition the optimization procedure into two phases and analyze the runtime of each phase. One difference is the calculation of the bound on the probability of generating a better offspring solution than a parent solution. The probability of each cell selected by the outer-level mutation in one round is $p = 1/(M-1)$. Thus, we have that the algorithm can decrease the distance $V_1$ (or $V_2$) by at least one with probability at least $p \cdot \gamma_{i,i+1} \cdot (1-p)^{M-1-1} \cdot |\boldsymbol{o}|_0$ (or $p \cdot \eta_{z,z'} \cdot (1-p)^{M-1-1} \cdot |\boldsymbol{o}|_0$). Then we can derive the one-step progress for each phase, and conclude that the expected runtime to find the target solution is $O(rM \ln(rM))$. The proof of Theorem 4.4 is similar to that of Theorem 4.2. The complete proofs of Theorems 4.3 and 4.4 are provided in **Appendix C.2** and **Appendix C.3**.

## 5. Experiments

In this section, we investigate the empirical performance of the (1+1)-ENAS algorithm with four different mutation settings for solving MCC. We set the problem classes $M$ from 2 to 24, with a step of 2, and the problem parameter $r$ from 2 to 10, with a step of 2. Figure 3 presents the average number of generations of 1,000 independent runs for finding an optimal solution. It shows that both local and global inner-level mutations can achieve similar performance. Furthermore, using one-bit or bit-wise outer-level mutations has little effect on performance, typically resulting in a constant factor difference. These empirical results are generally consistent with Theorems 4.1 to 4.4.

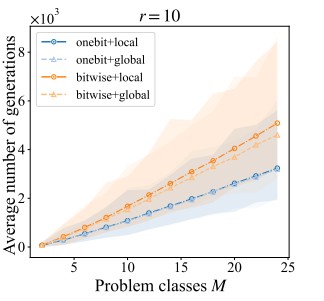
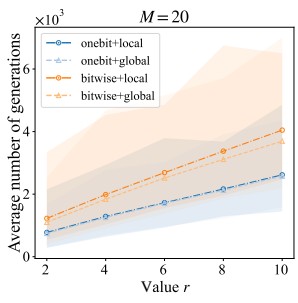

(a) Fixed $r = 10$, varying $M$    (b) Fixed $M = 20$, varying $r$

*Figure 3.* Average number of generations of the (1+1)-ENAS algorithm with different mutations for solving the MCC problem. The legend identifies mutation types: the term before the symbol "+" denotes the outer-level mutation, and the term after the symbol represents the inner-level mutation.

The above experiment results also offer practical guidance for mutation design in ENAS. In particular, many ENAS algorithms adopt bit-wise mutation (also called bit-flip mutation) (Xie & Yuille, 2017; Pan et al., 2024; Yan et al., 2024), which requires manual setting of an appropriate flipping probability; whereas the one-bit mutation, which is simpler in use, can be initially considered in the design of ENAS algorithms. This finding is particularly effective for block/cell-based search spaces, where one-bit mutation could achieve significant changes in neural architectures. This helps explain its adoption in practical ENAS methods like AE-CNN (Sun et al., 2019b), which utilize single-step mutation (functionally analogous to one-bit mutation) as their core search operator. To further strengthen our finding, we extend our experiments to population-based ENAS algorithms with crossover. Due to space limitations, the experiment results are included in **Appendix D**.

## 6. Conclusion

In this paper, we take the first step towards the runtime analysis of the ENAS algorithm for solving multiclass classification problems. We begin by proposing a multiclass classification problem MCC, and mathematically formulating a fitness function for this problem to serve as a benchmark for the runtime analysis of ENAS. Furthermore, we design a two-level search space with cells at the first level and blocks at the second, which is consistent with the practical ENAS algorithms and also enables theoretical research. Based on both designs, we analyze the expected runtime bounds of the (1+1)-ENAS algorithm with one-bit or bit-wise mutation for solving MCC, and prove that the algorithm using both mutations can achieve the same expected runtime bounds. The results suggest that a simple one-bit mutation can be initially considered in the ENAS community. As a foundation for the above theoretical research, the proposed benchmark MCC narrows the gap between previous theoretical analyses and practice.

Building on this work, more theoretical works on ENAS are left to be done, including but not limited to the following aspects: 1) analyzing more advanced evolutionary mechanisms/operators, such as population mechanisms, crossover operators, and stochastic selection mechanisms, to help the design of more efficient ENAS algorithms; 2) considering more realistic scenarios, such as noisy optimization, dynamic optimization, and multi-objective optimization; 3) extending to more challenging multi-class classification with richer decision boundaries, such as diverse polyhedra regions beyond sectors or triangles, to assess ENAS's capability in constructing blocks with diverse topologies.

## Impact statement

This paper presents work whose goal is to advance the field of Machine Learning. There are many potential societal consequences of our work, none which we feel must be specifically highlighted here.

## Acknowledgments

The authors want to thank the anonymous reviewers for their helpful comments and suggestions. This work was supported by the National Natural Science Foundation of China (No. 62276175 and No. 62276124) and Innovative Research Group Program of Natural Science Foundation of Sichuan Province (No. 2024NSFTD0035).

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

# A. Neural Architecture

## A.1. Details of Parameter Settings

**Parameters of the binary neurons within the block.** The weight $\varphi$ and bias of a binary neuron corresponds to the angle of the unit normal vector for the hyperplane and the distance from the origin, respectively (Fischer et al., 2023). We assume that the parameters, i.e., the weight $\varphi$ and bias, satisfy the following conditions:

- The bias of the neuron in an A-type block is set as $\cos(\pi/n)$.

- The bias of the neurons in a B-type block is set as 0, and the difference in $\varphi$ between these neurons is $(\pi + 2\pi/n)$ or $(\pi - 2\pi/n)$.

- Two neurons in a C-type block have a bias of 0, with $\varphi$ difference of $(\pi + 2\pi/n)$ or $(\pi - 2\pi/n)$, while the third neuron has a bias of $\cos(\pi/n)$ and a $\varphi$ difference with the other two neurons of $(\pi/2 + \pi/n)$ or $(3\pi/2 - \pi/n)$.

- Existing an optimization method can theoretically guarantee the optimality of the remaining parameters, e.g., $\varphi$ of each binary neuron and $\varphi$ difference.

Parameters for each type of block are depicted in Figure 4.

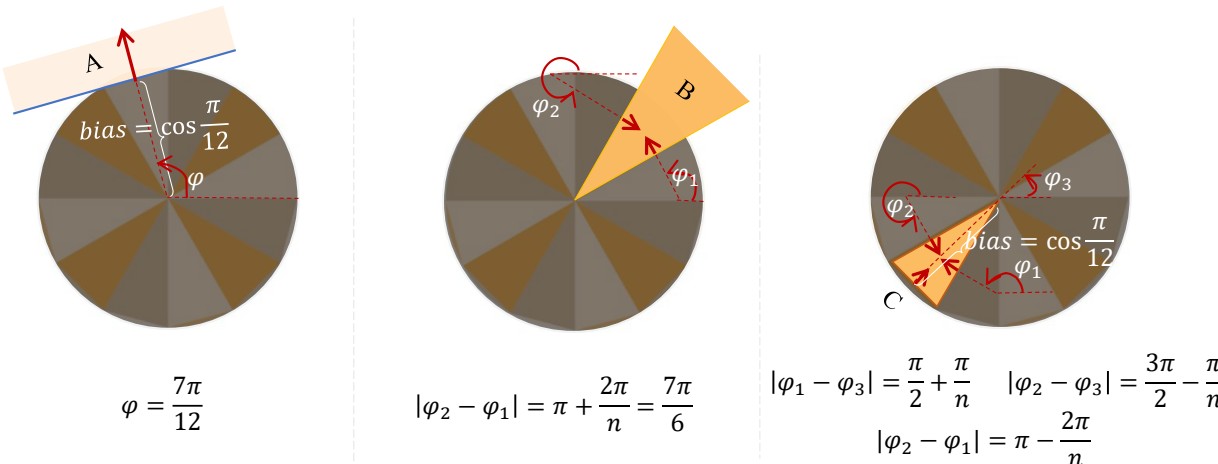

$$\varphi = \frac{7\pi}{12}$$

$$|\varphi_2 - \varphi_1| = \pi + \frac{2\pi}{n} = \frac{7\pi}{6}$$

$$|\varphi_1 - \varphi_3| = \frac{\pi}{2} + \frac{\pi}{n} \quad |\varphi_2 - \varphi_3| = \frac{3\pi}{2} - \frac{\pi}{n}$$

$$|\varphi_2 - \varphi_1| = \pi - \frac{2\pi}{n}$$

*Figure 4.* Visualization of the parameter settings for A-type, B-type, and C-type blocks, along with the corresponding decision regions generated by each block type, upon the Mcc problem ($M = 3, r = 2, n = 12$). The red arrow represents the unit normal vector of a hyperplane (line), with weight $\varphi$ as its angle. The bias is the distance of the hyperplane to the circle's center. Each block's arrows collectively point to the region produced (covered/bounds) by that block. The black-shaded regions are incorrectly classified by the specific block. (a) Top-left: The decision region (a segment region $S_{\text{sec}}^4$) produced by an A-type block with its single binary neuron's bias of $\cos(\pi/n)$ and $\varphi$ of $7\pi/12$. (b) Top-right: The decision region (a sector region $S_{\text{sec}}^2$) produced by a B-type block, where both neurons have a bias of 0, and the $\varphi$ difference between the neurons is $|\varphi_2 - \varphi_1| = \pi + 2\pi/n = 7\pi/6$. This decision region is generated by (c) Bottom: The decision region (a triangle region $S_{\text{tri}}^8$) produced by a C-type block, where the two binary neurons, each with a bias of 0, have a $\varphi$ difference of $|\varphi_2 - \varphi_1| = \pi - 2\pi/n = 5\pi/6$, and their $\varphi$ differences with the third binary are $|\varphi_1 - \varphi_3| = \pi/2 + \pi/n = 7\pi/12$ and $|\varphi_2 - \varphi_3| = 3\pi/2 - \pi/n = 17\pi/12$, respectively.

**Parameters of the neurons $\{N_1, N_2, \ldots, N_M\}$ in hidden layer.** Figure 5 shows the weights and biases of these neurons. Specifically, neuron $N_1$ has two inputs with weights of $\{1, 0.4\}$, neuron $N_M$ has a single input with a weight of 0.5, and each of the remaining neurons (i.e., $N_2, ..., N_{M-1}$) has three inputs with weights of $\{0.5, 1, 0.4\}$. Additionally, the bias for neuron $N_M$ is 0.1, whereas all other neurons in $\{N_1, N_2, \ldots, N_M\}$ have a bias of 0.

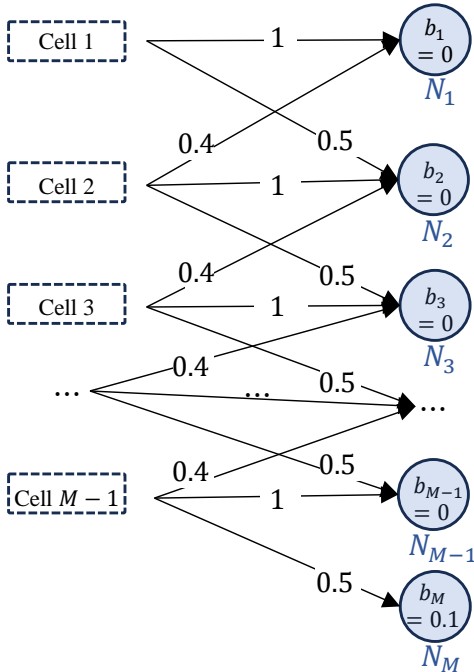

*Figure 5.* Parameters (weight and bias) of the neurons in $\{N_1, N_2, \ldots, N_M\}$.

## A.2. Decision Regions

Based on the above parameter settings, it is clear that when the dataset is on the unit circle, three types of blocks in the neural architecture generate decision regions with distinct shapes. Specifically, an A-type block in the neural architecture produces a sector-shaped decision region, as shown in the top-left example of Figure 4; a B-type block produces a sector-shaped decision region, as shown in the top-right example of Figure 4; and a C-type block produces a triangle-shaped decision region, as shown in the bottom example of Figure 4. Overall, A-type blocks can only produce the segments (including the segments within the sectors), B-type blocks can produce the triangles and sectors, and C-type blocks can produce the triangles (including the triangles within the sectors).

In addition, since the utilization of hidden layers (i.e., neurons $\{N_1, N_2, \ldots, N_M\}$), each block in the $m$-th cell will only form decision regions belonging to class $m$, except for the segment region that might be incorrectly classified as class $(M + m - 1) \mod M$. This exception occurs because, in the $((M + m - 1) \mod M)$-th cell, a B-type block is used to form the triangle region $S_{\mathrm{tri}}^i \in \mathrm{Tri}^{(M+m-1) \mod M}$ instead of a C-type block, which leads to the incorrect coverage of the segment region $S_{\mathrm{seg}}^i$ connected with $S_{\mathrm{tri}}^i$, where $S_{\mathrm{seg}}^i$ belongs to class $m$ (according to the third characteristic of MCC shown in Section 3.1 of the main paper). However, once an A-type block in the $m$-th cell is used to cover $S_{\mathrm{seg}}^i$, the neural architecture can ultimately classify all of the points in $S_{\mathrm{seg}}^i$ correctly. For further discussion on how the neural network classifies correctly in this case, please refer to the classification examples provided in **Appendix B**, particularly for the case where the input instance is $x \in S_{\mathrm{seg}}^7$.

## B. Examples for Neural Architecture Solving MCC

To detail how the neural architecture solves MCC, we utilize the example of MCC ($M = 3$ and $r = 2$) in the main paper to give an explanation. Table 1 illustrates several input instances along with the corresponding classification process and results. Specifically, this table includes the outputs of cells, neurons in $\{N_1, N_2, \ldots, N_M\}$, softmax layer, the final classification results, and whether classifications are right or not for seven input instances. Rows 1–2 present the classification process and results for input instance labeled as class 1, with three regions whose points will be misclassified; rows 3–5 show the classification process and results for input instance labeled as class 2, with two regions misclassified; rows 6–7 display

*Table 1.* Examples for testing the performance of neural architecture $\{(0,3,0),(1,3,0)\}$, including classification tests for seven input instances in MCC ($M = 3, r = 2$). The first column lists the case number, while the second and third columns display the seven input instances and their corresponding labels, respectively. Columns 4–7 provide the outputs from cells, neurons in $\{N_1, N_2, \ldots, N_M\}$, softmax, and the final classification results. The bolded numbers in columns 5 and 6 represent the maximum values among its output. For each case, the output of the softmax layer is a set of probabilities (i.e., $\{P_1, P_2, P_3\}$), with the class corresponding to the maximum value being the classification result. Column 8 presents whether the neural architecture correctly classified the input instance; it shows "✓" if the classification is correct (i.e., its label and classification result are the same), and "✗" if it is not.

| | Input Instance $x$ | Label | Cells Output | $\{N_1, N_2, N_3\}$ Output | Softmax Layer Output | Classification Result | Right? |
|---|---|---|---|---|---|---|---|
| 1 | $x \in S_{\text{Sec}}^1 \cup S_{\text{Sec}}^4 \cup S_{\text{tri}}^7$ | Class 1 | 10 | $\{\mathbf{1.0}, 0.5, 0.1\}$ | $\{\mathbf{0.50}, 0.30, 0.20\}$ | Class 1 | ✓ |
| 2 | $x \in S_{\text{seg}}^9 \cup S_{\text{tri}}^{10} \cup S_{\text{seg}}^{12}$ | | 00 | $\{0.0, 0.0, \mathbf{0.1}\}$ | $\{0.32, 0.32, \mathbf{0.36}\}$ | Class 3 | ✗ |
| 3 | $x \in S_{\text{Sec}}^2 \cup S_{\text{Sec}}^5 \cup S_{\text{tri}}^{11}$ | | 01 | $\{0.4, \mathbf{1.0}, 0.6\}$ | $\{0.25, \mathbf{0.45}, 0.30\}$ | Class 2 | ✓ |
| 4 | $x \in S_{\text{tri}}^8 \cup S_{\text{seg}}^{10}$ | Class 2 | 00 | $\{0.0, 0.0, \mathbf{0.1}\}$ | $\{0.32, 0.32, \mathbf{0.36}\}$ | Class 3 | ✗ |
| 5 | $x \in S_{\text{seg}}^7$ | | 11 | $\{1.4, \mathbf{1.5}, 0.6\}$ | $\{0.39, \mathbf{0.43}, 0.18\}$ | Class 2 | ✓ |
| 6 | $x \in S_{\text{Sec}}^3 \cup S_{\text{Sec}}^6 \cup S_{\text{tri}}^9 \cup S_{\text{tri}}^{12} \cup S_{\text{seg}}^8$ | Class 3 | 00 | $\{0.0, 0.0, \mathbf{0.1}\}$ | $\{0.32, 0.32, \mathbf{0.36}\}$ | Class 3 | ✓ |
| 7 | $x \in S_{\text{seg}}^{11}$ | | 01 | $\{0.4, \mathbf{1.0}, 0.6\}$ | $\{0.25, \mathbf{0.45}, 0.30\}$ | Class 2 | ✗ |

the classification process and results for input instance labeled as class 3, with one region misclassified. The classification accuracy, calculated as the ratio of the area of correctly classified points to the area of the circle, is approximately 0.83.

## C. Proofs

### C.1. Proof of Theorem 4.2

*Proof of Theorem 4.2.* Similar to the proof of Theorem 4.1, we begin the proof by analyzing phase 1.

We first show that the expected number of 0-bits and 1-bits in the initial solution $x_0$ is $\mathbb{E}[|o_{x_0}|_0] = (M-1)(1-1/r^2)$ and $\mathbb{E}[|o_{x_0}|_1] = (M-1)/r^2$, respectively, where $o_{x_0}$ represents $o$ of $x_0$.

Let $P(o^m = 1)$ be the probability that the $m$-th cell of the initial solution has $I_{x_0}^m = 2r$ B-type and C-type blocks. Since both $n_B^m$ and $n_C^m$ drawn from the uniform distribution $U[1, r]$, the probability of $n_B^m + n_C^m = 2r$ (i.e., $o^m = 1$ for the initial solution since $I_{x_0}^m = 2r$) is $1/r^2$, and the probability of $n_B^m + n_C^m < 2r$ (i.e., $o_{x_0}^m = 0$) is $1 - 1/r^2$. Then, we have $P(o^m = 1) = 1/r^2$ and $P(o^m = 0) = 1 - 1/r^2$ for the initial solution. Thus, the expectation of $|o|_1$ for the initial solution is $(M-1)/r^2$.

By Markov's inequality, an initial solution has at least $g = (M-1)/r$ 0-bits in $o_{x_0}$ with probability

$$
\begin{aligned}
P\left(|o_{x_0}|_0 \geq g\right) &= P\left(|o_{x_0}|_1 < (M-1) - g\right) \\
&= 1 - P\left(|o_{x_0}|_1 \geq (M-1) - g\right) \\
&\geq 1 - \frac{\mathbb{E}[|o_{x_0}|_1]}{(M-1) - g} \\
&\geq 1 - \frac{(M-1)/r^2}{(M-1) - g} = 1 - \frac{1}{r^2 - r} \geq 1/2,
\end{aligned}
\tag{10}
$$

where the last inequality holds by $r \geq 2$.

Next, we show the expected runtime for finding a solution in subspace $\mathcal{S}_{n-2r}$, i.e.,

$$
\begin{aligned}
\mathbb{E}[T_1] &= \sum_{d=0}^{(M-1)-1} \mathbb{E}[T_1 \mid |\boldsymbol{o}_{\boldsymbol{x}_0}|_0 = d] \cdot P(|\boldsymbol{o}_{\boldsymbol{x}_0}|_0 = d) \\
&\geq \sum_{d=g}^{(M-1)-1} \mathbb{E}[T_1 \mid |\boldsymbol{o}_{\boldsymbol{x}_0}|_0 = d] \cdot P(|\boldsymbol{o}_{\boldsymbol{x}_0}|_0 = d) \\
&\geq \mathbb{E}[T_1 \mid |\boldsymbol{o}_{\boldsymbol{x}_0}|_0 = g] \cdot P(|\boldsymbol{o}_{\boldsymbol{x}_0}|_0 \geq g) \\
&= \mathbb{E}[T_1 \mid |\boldsymbol{o}_{\boldsymbol{x}_0}|_0 = g] \cdot 1/2 \geq t \cdot P(T_1 > t)/2,
\end{aligned}
\tag{11}
$$

where $P(T_1 > t)$ is the probability that the generations for finding a solution in $\mathcal{S}_{n-2r}$ is greater than $t$.

Let $G$ denote the event that at least one of these $g$ 0-bits in $\boldsymbol{o}_{\boldsymbol{x}_0}$ is never changed in $t = \max\{1, M-2\} \cdot \ln(M-1) \cdot r$. We show that the event $G$ happens with probability lower bounded by $1 - e^{-1}$. Let $1 - \gamma_{i,i+1}/(M-1)$ describe the probability that a new B or C-type block would not be generated for the cell with $o^m = 0$ in one round (i.e., value $I_{\boldsymbol{x}}^m$ is not increased), where $\gamma_{i,i+1}$ is a constant probability which detailed in the proof of Theorem 4.1. Then, the probability that the value $I_{\boldsymbol{x}}^m$ of a specific cell with $o^m = 0$ in the initial solution $\boldsymbol{x}_0$ is never increased in $t$ generations is $(1 - \gamma_{i,i+1}/(M-1))^t$. So, the value $I_{\boldsymbol{x}}^m$ of a specific cell with $o^m = 0$ in the initial solution $\boldsymbol{x}_0$ is increased by at least once in $t$ generations with a probability $1 - (1 - \gamma_{i,i+1}/(M-1))^t$. Furthermore, any of these cells corresponding to a 0-bit in $\boldsymbol{o}_{\boldsymbol{x}_0}$ is increased by one at least once in $t$ generations with a probability of $(1 - (1 - \gamma_{i,i+1}/(M-1))^t)^g$. Thus, we have

$$
\begin{aligned}
P(G) &= 1 - \left(1 - \left(1 - \frac{\gamma_{i,i+1}}{M-1}\right)^t\right)^g \\
&\geq 1 - \left(1 - \left(1 - \frac{1}{M-1}\right)^{\max\{1, M-2\} \cdot \ln(M-1) \cdot r}\right)^{\frac{M-1}{r}} \\
&\geq 1 - e^{-1}.
\end{aligned}
$$

Combining with Eq. (11), the lower bound of the expected runtime for the first phase can be derived by

$$
\begin{aligned}
\mathbb{E}[T_1] &\geq t \cdot P(T_1 > t)/2 \\
&\geq \max\{1, M-2\} \cdot \ln(M-1) \cdot r \cdot P(G)/2 \\
&\geq \max\{1, M-2\} \cdot \ln(M-1) \cdot r \cdot (1 - e^{-1})/2 \\
&\in \Omega(rM \ln M).
\end{aligned}
$$

Because the generations $T_1$ of phase 1 is a lower bound on the generations $T$ of the whole process for finding an optimal solution belonging to $\mathcal{S}_{n-2r}^{n-2r}$, we have $\mathbb{E}[T] = \Omega(rM \ln M)$. □

### C.2. Proof of Theorem 4.3

*Proof of Theorem 4.3.* Since the one-bit and bit-wise mutations have the same lower bound for the probability of increasing the value of $\mathbb{I}_{\boldsymbol{x}}$ (or $\mathbb{J}_{\boldsymbol{x}}$) by 1, we can obtain the same lower bound for the expected one-step progress (drift). Specifically, the drift of $V_1(\boldsymbol{x}_t)$ (or $V_2(\boldsymbol{x}_t)$) is bounded from below by $\sigma \cdot V_1(\boldsymbol{x}_t)$ (or $\sigma \cdot V_2(\boldsymbol{x}_t)$) with $\sigma = 2/(9N)$ (or $\sigma = 1/(9N)$). These results lead to the same upper bound results for the expected runtime, i.e., $O(rM \ln(rM))$, by utilizing the multiplicative drift analysis (Doerr et al., 2012). Therefore, a similar proof refers to the proof of Theorem 4.1 in the main paper.

Here, we present the proof with another approach that uses the fitness-level technique (Wegener, 2003; Sudholt, 2013), which also yields the same theoretical result.

Let $p_I$ denote the lower bound on the probability of the algorithm creating a new solution in $\bigcup_{I'=I+1}^{n-2r} \mathcal{S}_{I'}$, provided the algorithm is in $\mathcal{S}_I$. The most optimistic assumption is that the algorithm makes it rise one level (i.e., $\mathcal{S}_{I+1}$). To make jumping from $\mathcal{S}_I$ to $\mathcal{S}_{I+1}$ successful, it is sufficient to select one 0-bit (denote as the $m$-th cell) from $\boldsymbol{o}$ by bit-wise mutation and mutate the $m$-th cell to increase its value $I_{\boldsymbol{x}}^m$ by one through inner-level mutation, which happens with a probability

of $\frac{1}{M-1} \cdot |\boldsymbol{o}|_0 \cdot \gamma_{i,i+1}$. Meanwhile, the other 0 and 1 bits are either not selected or their corresponding $I_{\boldsymbol{x}}^m$ values remain unchanged after being mutated, which happens with a probability of at least $(1 - \frac{1}{M-1})^{(M-1)-1} \geq 1/e$. Thus, we have

$$
p_I \geq \frac{\gamma_{i,i+1}}{M-1} \cdot |\boldsymbol{o}|_0 \cdot \frac{1}{e} \geq \left(1 - \frac{I}{N}\right) \cdot \frac{2}{9e},
$$

where the last inequality holds by $|\boldsymbol{o}|_0 \geq M - 1 - I/(2r)$ (derived by Eq. (3) in the main paper), $N = n - 2r = 2r(M-1)$, and $\gamma_{i,i+1} \geq 2/9$. Thus, the expected runtime for the first phase is

$$
\begin{aligned}
\mathbb{E}[T_1] &\leq \sum_{I=0}^{N-1} \frac{1}{p_I} \leq \frac{9e}{2} \cdot \sum_{I=0}^{N-1} \frac{1}{1 - I/N} \\
&\leq \frac{9e}{2} \cdot N \sum_{I=0}^{N-1} \frac{1}{N-I} \leq \frac{9e}{2} N \ln N \in O\left(rM \ln(rM)\right).
\end{aligned}
$$

Then, we analyze the second phase. Before beginning the analysis, it is important to note that $\mathbb{J}_{\boldsymbol{x}} - \epsilon_{\boldsymbol{x}} \geq 0$ for any solution $\boldsymbol{x}$. This is because $\epsilon_{\boldsymbol{x}}$ will only be positive if the $(M-1)$-th cell contains more than $r$ B-type blocks (leading to $\mathbb{J}_{\boldsymbol{x}} \geq r$ since $J_{\boldsymbol{x}}^{M-1} \geq r$), which allows for $n_B'' > 0$ B-type blocks to cover triangles belonging to class $M-1$, making $\epsilon_{\boldsymbol{x}} > 0$. In this case, we have $\mathbb{J}_{\boldsymbol{x}} - \epsilon_{\boldsymbol{x}} \geq 0$ since $\mathbb{J}_{\boldsymbol{x}} \geq r$ and $\epsilon_{\boldsymbol{x}} \leq r$. In all other cases, where $\epsilon_{\boldsymbol{x}} = 0$, we have $\mathbb{J}_{\boldsymbol{x}} - \epsilon_{\boldsymbol{x}} \geq 0$ since $\mathbb{J}_{\boldsymbol{x}} \geq 0$. Therefore, we have $\mathbb{J}_{\boldsymbol{x}} - \epsilon_{\boldsymbol{x}} = J - r \geq 0$, given that the algorithm is in $\mathcal{S}_N^J$ (i.e., $\boldsymbol{x} \in \mathcal{S}_N^J$).

Next, we continue the analysis of the runtime for the second phase. Since $\eta_{z,z+1} \geq 1/9$ as shown in the proof of Theorem 4.1, we can derive that the lower bound on the probability of jumping from $\mathcal{S}_N^J$ to $\mathcal{S}_N^{J+1}$ is $\eta_{z,z+1}/(M-1) \cdot |\boldsymbol{o}|_0/e \geq (1 - (J-r)/N)/(9e)$, where $|\boldsymbol{o}|_0 = M - 1 - |\boldsymbol{o}|_1 \geq M - 1 - (J-r)/(2r)$ and $J - r \geq 0$. Then, similar to the first phase, we have the expected runtime for the second phase is $\mathbb{E}[T_2] \in O(rM \ln(rM))$.

By combining the two phases, we get $\mathbb{E}[T] \in O\left(rM \ln(rM)\right)$. $\qquad\square$

### C.3. Proof of Theorem 4.4

*Proof of Theorem 4.4.* Similar to the proof of Theorem 4.2, the probability of the event that the algorithm generates an initial individual $\boldsymbol{x}_0$ with $(M-1)/r$ 0-bits in $\boldsymbol{o}$ is lower bounded by $1/2$. For a specific cell with $o^m = 0$, the probability of the event that the number of B-type and C-type blocks in this cell increased is bounded by $\sum_{i'=i+1}^{2r} \gamma_{i,i'}/(M-1) \leq 1/(M-1)$. Then, the proof follows a similar approach to that of Theorem 4.2. The event that at least one of the cells with $o^m = 0$ in $\boldsymbol{x}_0$ is never changed in $t = \max\{1, M-2\} \cdot \ln(M-1) \cdot r$ generations happens with probability exceeding $(1 - e^{-1})$. Thus, the probability of the event that the generations for finding a solution in $\mathcal{S}_{n-2r}$ is greater than $t$ is lower bounded by $(1 - e^{-1})$. Starting from the initial individual $\boldsymbol{x}_0$, the algorithm finds a solution in subspace $\mathcal{S}_{n-2r}$ is $O(rM \ln M)$ according to Eq. (11). Because the generations of phase 1 is a lower bound on the generations $T$ of the whole process for finding an optimal solution belonging to $\mathcal{S}_{n-2r}^{n-2r}$, we have $\mathbb{E}[T] = \Omega(rM \ln M)$. $\qquad\square$

## D. Extended Experiments

To further strengthen our findings, we extend the experiments in Section 5 to three SOTA ENAS algorithms: $(\lambda+\lambda)$-ENAS with mutation only ($\lambda \in \{2, 4, 10\}$), which is adopted in methods like LEIC (Real et al., 2017) and AmoebaNet (Real et al., 2019); one-point crossover-based ENAS, which is adopted in CNN-GA (Sun et al., 2019a) and ENAS-kT (Yang et al., 2023); and uniform crossover-based ENAS, which is adopted in Genetic CNN (Xie & Yuille, 2017). All experiments follow the same problem settings as in Section 5, and the results are presented in Figure 6 and Figure 7. The results show that one-bit mutation achieves comparable runtime performance to bit-wise mutation on the MCC problem.

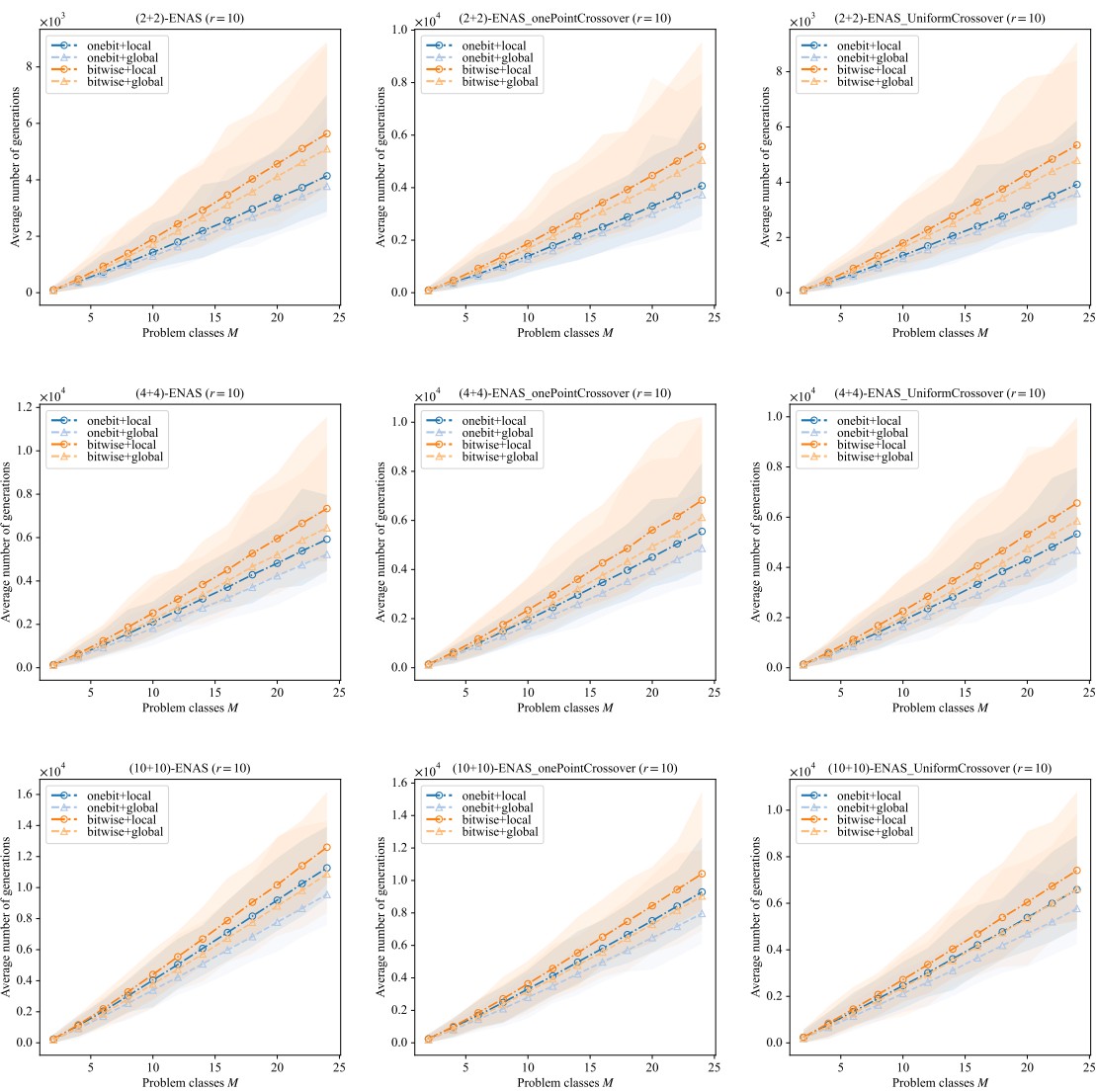

*Figure 6.* Average number of generations of the ENAS algorithms with population and crossover for solving the MCC problem ($r = 10$, varying $M$). The ENAS algorithms include ($\lambda+\lambda$)-ENAS algorithm (with mutation only), ($\lambda+\lambda$)-ENAS algorithm with one-point crossover, and ($\lambda+\lambda$)-ENAS algorithm with uniform crossover, where $\lambda \in \{2, 4, 10\}$.

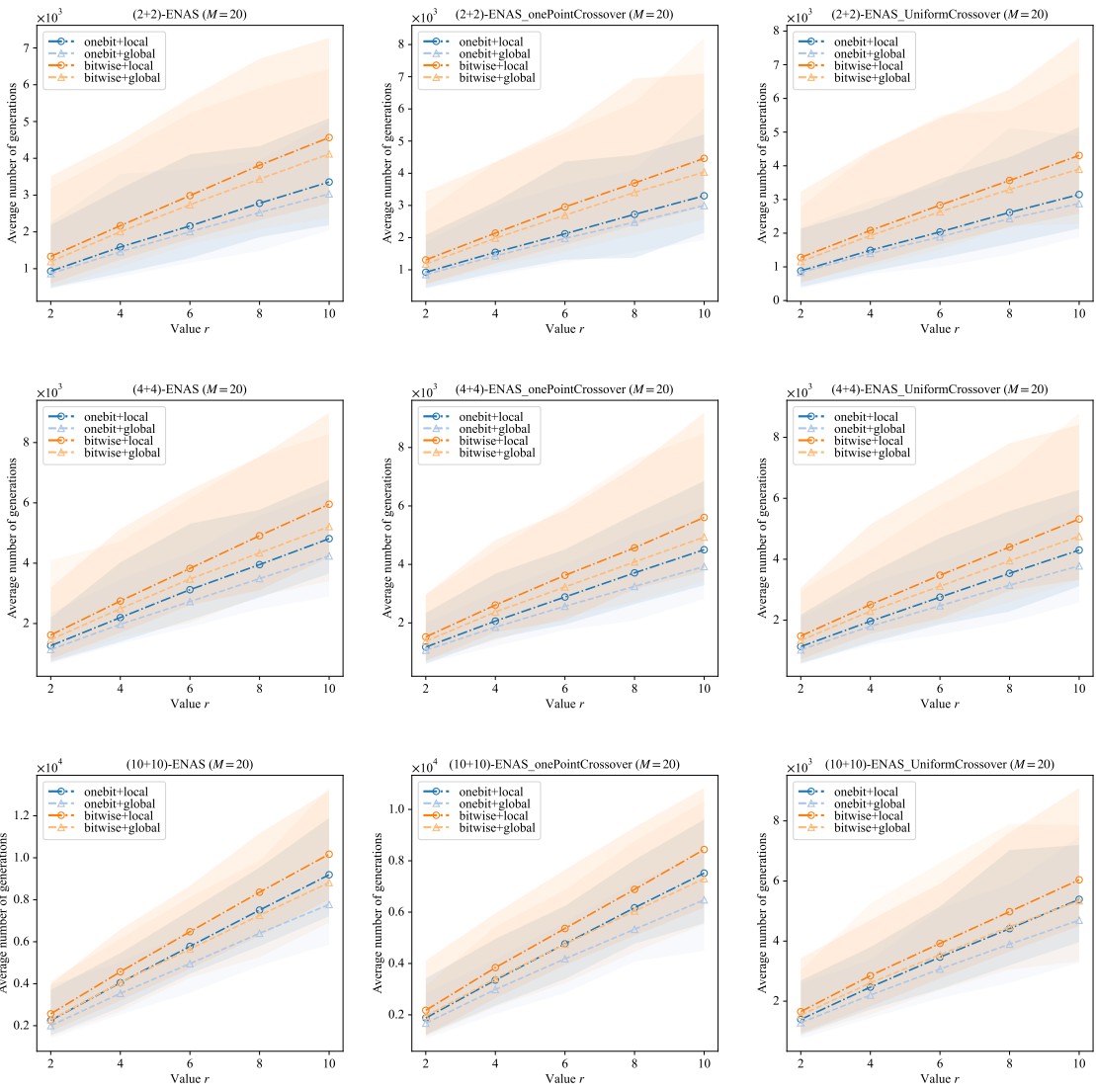

*Figure 7.* Average number of generations of the ENAS algorithms with population and crossover for solving the MCC problem ($M = 20$, varying $r$). The ENAS algorithms include ($\lambda$+$\lambda$)-ENAS algorithm (with mutation only), ($\lambda$+$\lambda$)-ENAS algorithm with one-point crossover, and ($\lambda$+$\lambda$)-ENAS algorithm with uniform crossover, where $\lambda \in \{2, 4, 10\}$.

