# OpenReview forum: "Runtime Analysis of Evolutionary NAS for Multiclass Classification"
_ICML.cc/2025/Conference — ICML 2025 poster_

### Official Review · Reviewer_26fJ · 2025-03-10

**Overall Recommendation:** 4

**Summary:**

This study presents a theoretical analysis of the runtime of the ENAS algorithm in solving multiclass classification problems. The authors first introduce a benchmark problem and then propose a two-level search space. Based on this design, the authors analyze the upper and lower bounds on the expected runtime and provide empirical results that validate their theoretical findings. The findings indicate that the simple one-bit mutation can achieve similar performance to the commonly used bit-wise mutation of ENAS.

**Claims And Evidence:**

All the claims of this study are well supported.

**Essential References Not Discussed:**

This study focuses on the runtime analysis of the ENAS algorithm, and the introduction and related works have discussed the more relevant aspects of the topic.

**Experimental Designs Or Analyses:**

The experiments can justify the claims of this study, but more datasets should be better.

**Methods And Evaluation Criteria:**

Both the introduced benchmark problem and the two-level search space can well support the ENAS problem at hand.

**Other Comments Or Suggestions:**

Here are some minor weaknesses of this study:
1. This study focuses only on the mutation operations of the ENAS algorithm, ignoring other important evolutionary operators, e.g., crossover and selection operations, which are commonly used in practical ENAS implementations. The authors should discuss them in the limitation section.
2. The experiments of this study are limited in scope. Including more datasets or SOTA ENAS algorithm could strengthen the findings.

**Other Strengths And Weaknesses:**

Strengths
1. The paper provides the important theoretical findings of ENAS. The findings are crucial, especially for the multiclass optimization task for ENAS.
2. The insights of the mutation operations can guide the design of the ENAS algorithms and also can simplify the existing ENAS algorithms.
3. The upper and lower bounds for the runtime of ENAS, which can effectively guide the mutation selection of the ENAS algorithm.
4. The definition of the MCC benchmark problem and the fitness function are good to support the following theoretical proof.
5. The two-level search space aligns well with the application of ENAS algorithms in practice, making the theoretical findings applicable to real-world ENAS settings.
6. All the theorems of the paper, such as runtime bounds and fitness function, are proved correctly.
7. The empirical analysis is designed to verify the effectiveness of the theoretical findings.
8. The paper is well-organized and the presentation is easy to understand.

Weaknesses
1. This study analyzes the (1+1)-ENAS algorithm and finds that simple one-bit mutation can achieve similar performance as commonly used bit-wise mutation. However, are the ENAS algorithms widely used in practice are (1+1)-ENAS? In other words, can this finding well support the widely used ENAS algorithms in practice?
2. The differences and the challenges of the multiclass classification and binary classification for ENAS algorithms should be well-defined. Based on this definition, the contribution of this study can be more meaningful.
3. Although the authors justify that one-bit mutation can be a powerful mutation operation for ENAS, they do not provide strong practical guidelines on how practitioners should choose between one-bit and bit-wise mutations in different settings. Do the authors believe that all the bit-wise mutations in ENAS should be replaced with one-bit mutation?

**Questions For Authors:**

Please check Other Strengths And Weaknesses and Other Comments Or Suggestion.

**Relation To Broader Scientific Literature:**

The paper contributes to the theoretical understanding of ENAS by analyzing runtime bounds, which can be regarded as a continual work of Lv et al. This work extended the study from binary classification tasks to multclass classification tasks.

Lv Z, Bian C, Qian C, et al. Runtime Analysis of Population-based Evolutionary Neural Architecture Search for a Binary Classification Problem[C]//Proceedings of the Genetic and Evolutionary Computation Conference. 2024: 358-366.

**Theoretical Claims:**

I have checked the correctness of the proofs for theoretical claims in this work, including the fitness function, the upper bound, and the lower bound, which are well designed and proofed.

---

> ### Author Rebuttal · Authors · 2025-04-01
>
> Many thanks for your valuable comments of our work. Below, we would like to take this opportunity to respond to your concerns.
>
> >1. Are the ENAS algorithms widely used in practice are (1+1)-ENAS? In other words, can this finding well support the widely used ENAS algorithms in practice?
>
> (1) In practice, most ENAS algorithms are population-based, while (1+1)-ENAS serves as a theoretical foundation for runtime analysis and designing population-based algorithms. (2) Yes. Our finding shows that simple one-bit mutation achieves comparable runtime performance to bit-wise mutation on the MCC problem, validating that the one-bit mutation is not inherently inefficient. This aligns with practical ENAS methods like AmoebaNet (Real et al., 2019) and AE-CNN (Sun et al., 2019b), which utilize single-step mutation (functionally analogous to one-bit mutation) as their core search operator. Thanks to your comment, we will revise to discuss the connection to practical implementation.
>
> >2.The differences and the challenges of the multiclass classification and binary classification for ENAS algorithms should be well-defined.
>
> Differences: 1) **decision regions**: multiclass classification divides input space into $M$ decision regions (vs. two in binary classification), increasing neural architectural demands; 2) **classification accuracy**: multiclass classification aggregates per-class accuracy across all $M$ regions, amplifying the complexity of the fitness evaluation; 3) **search space**: the neural architecture for solving multiclass classification is a combination of multiple binary classifiers or a more complex architecture (vs. binary classification’s binary classifier), exponentially expanding the search space.
> Challenges: 1) **problem definition**: accurately modeling inter-class dependencies and region-specific sample distributions; 2) **fitness function**: mathematically formulating the fitness (i.e., classification accuracy) of neural architecture from geometric properties; 3) **search space partition**: partitioning the search space by analyzing the interactions between architectural components (e.g., blocks, cells).
> Thanks to your suggestion, we will incorporate these clarifications into the revised manuscript.
>
> >3. Do the authors believe that all the bit-wise mutations in ENAS should be replaced with one-bit mutation?
>
> No. Consistent with the No Free Lunch Theorem, neither operator is universally optimal: bit-wise mutation excels in exploration-heavy scenarios (e.g., topology-based search spaces), while one-bit mutation prioritizes simplicity and serves as a parameter-free alternative (e.g., no need to tune bit-flip probability). As an initial attempt at ENAS runtime analysis, we prioritize foundational cases, while future work will expand comparisons to diverse tasks. We appreciate your support as we advance this work.
>
> >4.This study focuses only on mutation, ignoring other important evolutionary operators, which should be noted as a limitation.
>
> Our study focuses on mutation as it is a core component of ENAS and many ENAS are limited to mutation only (Real et al., 2017; Real et al., 2019; So et al., 2021). This also aligns with the theory community of evolutionary computation, where initial analyses often start with mutation (Auger & Doerr, 2011; Neumann & Witt, 2010; Zhou et al., 2019; Doerr & Neumann, 2020), thereby providing a stepping stone for further exploration. While advanced operators like crossover and stochastic selection are vital in practice, their runtime analysis remains challenging due to irregular combinatorial interactions and limited theoretical frameworks. We will explicitly discuss this.
>
> >5.The experiments of this study are limited in scope. Including more datasets or SOTA ENAS algorithm could strengthen the findings.
>
> Our search space is simplified for the MCC problem, which makes it difficult to directly apply the studied ENAS framework to other domains (e.g., image classification) without architectural re-engineering. Future work will expand this framework to diverse tasks. In addition, to further strengthen our finding, we have extended experiments to three SOTA ENAS algorithms: ($\lambda$+$\lambda$)-ENAS with mutation only ($\lambda\in\{2,4,10\}$), which is adopted in methods like LEIC (Real et al., 2017) and AmoebaNet (Real et al., 2019); one-point crossover-based ENAS, which is adopted in CNN-GA (Sun et al., 2019a) and ENAS-kT (Yang et al., 2023); and uniform crossover-based ENAS, which is adopted in Genetic CNN (Xie & Yuille, 2017). All additional experiments are under identical problem settings (Section 5). The results (see anonymous links: [fixed $r=10$, varing $M$](https://anonymous.4open.science/r/5666/rebuttalFigs/a-varing-M.pdf) and [Fixed $M = 20$, varying $r$](https://anonymous.4open.science/r/5666/rebuttalFigs/b-varing-r.pdf)) show that one-bit mutation still performs better. Thanks to your suggestion, we will integrate them into the manuscript.

---

### Official Review · Reviewer_iu6X · 2025-03-11

**Overall Recommendation:** 4

**Summary:**

This paper introduces a novel runtime analysis framework for evolutionary neural architecture search (ENAS). Compared with the previous studies, which focus on the binary classification problem, the runtime analysis focuses on the multiclass classification problem in this work. Specifically, this study first introduces a multiclass classification benchmark problem called MCC. Then, a more practical search space with two interrelated levels is designed. Based on this, the expected runtime bounds of (1+1)-ENAS are analyzed. ## update after rebuttal

**Claims And Evidence:**

All the claims made in this work are well supported by theoretical proofs or experimental results, none of which is problematic.

**Essential References Not Discussed:**

None.

**Experimental Designs Or Analyses:**

I have checked the experimental results regarding the different mutation settings in (1+1)-ENAS algorithm. This set of results can well support the claims made in Theorems 4.1 to 4.4.

**Methods And Evaluation Criteria:**

The proposed method and evaluation criteria indeed make sense for the problem in this work. The multiclass classification benchmark problem successfully extends the runtime analysis from binary classification to multiclass classification.

**Other Comments Or Suggestions:**

None.

**Other Strengths And Weaknesses:**

Strengths:
+ This paper mainly focuses on the runtime analysis of ENAS, which is a very important topic in NAS research. The authors provided several novel viewpoints in this field.
+ This paper pushes the runtime analysis of ENAS toward the new area regarding the multiclass classification problem. Compared with the previous studies focusing on the binary classification problem, this work achieves significant improvements upon them.
+ A multiclass classification benchmark problem and a novel fitness function are introduced. The fitness function not only contributes to the theoretical analysis of ENAS, but also can be transferred to other fields in evolutionary computation.
+ The authors present a practical search space specifically for the multiclass classification problem, pushing the search space design towards a new direction in the field of ENAS.
+ All the claims in this work are well supported by sufficient theoretical proofs or experimental results.
+ The expected runtime bounds obtained contribute to the design of ENAS algorithms, especially for the design of the mutation operator.
+ The paper is very well presented, and the illustrations and visualizations are nicely done.

Weaknesses:
+ The authors state that the expected runtime to find the target solution is $O(rMln(rM))$. However, it is suggested to discuss how this conclusion contributes to ENAS.
+ I wonder if the theoretical findings are valid when there are infinite classes. Please provide more details regarding this point.
+ Is there a sequence for the two types of mutations in the two-level mutation process? How does this sequence affect the theoretical analysis in this work?
+ In lines 223-229 of page 5, the authors state that the definition of classification accuracy is suitable for balanced datasets. I wonder if the distribution of data is imbalanced (e.g., long-tailed data), what impact will it have on the theoretical findings?

**Questions For Authors:**

Please refer to the Other Strengths And Weaknesses section.

**Relation To Broader Scientific Literature:**

In terms of the runtime analysis for ENAS, previous studies mainly focus on the binary classification problem, and their representation of the search space limits the application to multiclass classification problem. Different from these studies, this work achieves the analysis for multiclass problem via the proposed multiclass classification benchmark problem and the search space with two inter-related levels.

**Theoretical Claims:**

I have checked the correctness of all proofs and theoretical claims, and believe that they are all correct.

---

> ### Author Rebuttal · Authors · 2025-04-01
>
> Many thanks for your recognition and comments of our work. Below, we would like to take this opportunity to address your concerns.
>
> >1. The authors state that the expected runtime to find the target solution is $O(rM\ln(rM))$. However, it is suggested to discuss how this conclusion contributes to ENAS.
>
> This result highlights the relationship between the expected runtime and key problem parameters, i.e., $M$ (the number of classes) and $r$ (which affects the decision regions for each class). The result provides two insights for ENAS: 1) due to the polynomial runtime dependency on classification problem parameters $M$ and $r$, the ENAS generations (iterations) should scale with classification difficulty; 2) since our runtime results (Theorems 4.1 to 4.4) show that one-bit and bit-wise mutation achieve comparable runtime in (1+1)-ENAS solving MCC problem, the one-bit mutation as a simpler and parameter-free operator (e.g., no need to tune bit-flip probability) can be prioritized in ENAS design.
>
> >2. I wonder if the theoretical findings are valid when there are infinite classes. Please provide more details regarding this point.
>
> Yes. As shown in Theorems 4.1 to 4.4, the expected runtime $\mathbb{E}[T]$ holds with any number $M$ of classes. This finding shows that the architecture search difficulty increases with $M$ due to the finer partitioning of the decision spaces. Moreover, the theoretical framework does not impose any upper limit on the number $M$ of classes, meaning the findings apply regardless of whether $M$ is finite or tends to infinity. However, in extreme cases where $M$ becomes infinitely large, practical considerations such as finite computational resources should be taken into account.
>
> >3. Is there a sequence for the two types of mutations in the two-level mutation process? How does this sequence affect the theoretical analysis in this work?
>
> Yes, there is a sequence in the two-level mutation process. By first selecting the cell to mutate (outer-level mutation) and then applying the mutation (inner-level mutation) to the selected cell, we can track the expected changes between the parent and offspring, and also can quantify the progress made at each step. However, reversing the sequence would make the algorithm infeasible, as the index of the cell to be mutated would be unknown, making it unclear where the inner-level mutation should be applied. Specifically, given a solution $\pmb{x}$, which is encoded by $M-1$ triplets of integers, i.e., $\pmb{x} =${$(n_A^1, n_B^1, n_C^1),\ldots,(n_A^{M-1}, n_B^{M-1}, n_C^{M-1})$},  the algorithm must first select the cell index $m\in${$1,\ldots,M-1$}, and then apply the inner-level mutation to modify the selected cell $(n_A^m,n_B^m,n_C^m)$.
>
> >4. In lines 223-229 of page 5, the authors state that the definition of classification accuracy is suitable for balanced datasets. I wonder if the distribution of data is imbalanced (e.g., long-tailed data), what impact will it have on the theoretical findings?
>
> If the data distribution is imbalanced, the formulation of the fitness function (i.e., Eq. (2)) will need to consider the distribution of each class of data, which will introduce new conditions for judging whether each cell reaches the optimal state or not in the ENAS search process, e.g., the conditions $I_{\pmb{x}}^m=2r$ and $J_{\pmb{x}}^m=2r$ for $o^m$ to be 1 (as stated in line 353) will be affected by the distribution of data. Furthermore, a dependency would exist between the number of optimal cells $|\pmb{o}|_1$ in Eq. (3) and the distribution of data. Consequently, this affects the computation of expected progress (e.g., Eq. (4)) and the distribution of the optimal cells in the initial solution (e.g., Eq. (6)), ultimately making the theoretical findings (i.e., expected runtime shown in Theorems 4.1– 4.4) dependent on the data distribution.

---

### Official Review · Reviewer_dy7r · 2025-03-14

**Overall Recommendation:** 3

**Summary:**

This paper investigates the runtime analysis of Evolutionary Neural Architecture Search for multiclass classification problems. The core of the research is the proposal of a multiclass classification benchmark problem and the design of a two-level search space based on this problem. The authors then analyze the expected runtime bounds of the (1+1) ENAS algorithm with one-bit and bit-wise mutations for solving MCC. The results show that both mutation strategies achieve similar performance in terms of expected runtime, which is further verified by empirical studies. This work represents the first theoretical analysis of ENAS for multiclass classification, providing new insights into understanding ENAS.

## update after rebuttal

After the rebuttal, I feel my concerns have been well-discussed by the authors, although some of weaknesses are not very easy to modify during a such short period. Therefore, I choose to keep my positive scores, and I believe that all the reviewers are lean towards to accept this submission.

**Claims And Evidence:**

The claims made in the submission are generally supported by clear and convincing evidence, particularly in the context of the theoretical analysis and the proposed benchmark problem.

**Essential References Not Discussed:**

There is no essential references missing.

**Experimental Designs Or Analyses:**

Yes, I reviewed the soundness and validity of the experimental design and analysis presented in the paper, focusing on the empirical evaluation of the (1+1)-ENAS algorithm with different mutation strategies on the proposed MCC problem.

**Methods And Evaluation Criteria:**

The proposed methods and evaluation criteria in this paper are well-aligned with the problem of runtime analysis for evolutionary neural architecture search in multiclass classification.

**Other Comments Or Suggestions:**

Overall, the manuscript is technically sound and well-presented, with some theoretical insights. Please refer to other strengths and weaknesses for drawbacks.

**Other Strengths And Weaknesses:**

**Strengths,**
- This paper is the first to conduct a runtime analysis of ENAS for multiclass classification problems. By proposing the MCC problem and a mathematically formulated fitness function, it provides a benchmark for theoretical research on ENAS.
- The authors use rigorous mathematical tools, such as multiplicative drift analysis and fitness-level techniques, to derive upper and lower bounds on the expected runtime.
- The findings suggest that simpler mutation strategies like one-bit mutation can be effective in ENAS, which may simplify the design of future ENAS algorithms.

**Weaknesses,**
- How does the proposed runtime analysis scale with increasing problem complexity (e.g., higher-dimensional input spaces or larger numbers of classes)?  Are there any limitations in extending the analysis to more complex scenarios?
- The fitness function assumes optimal parameter tuning for each architecture. How sensitive are the theoretical results to deviations from this assumption?  Could the analysis be extended to account for suboptimal parameter tuning?
- The paper shows that one-bit and bit-wise mutations achieve similar performance. How do these findings influence the design of future ENAS algorithms?
- Can the theoretical framework be extended to analyze more complex ENAS algorithms? If so, what are the key challenges?

**Questions For Authors:**

Please refer to my weaknesses listed above.

**Relation To Broader Scientific Literature:**

The key contributions of this paper are related to several important areas of the broader scientific literature, particularly in the fields of Evolutionary Computation, Neural Architecture Search, and Runtime Analysis of Evolutionary Algorithms.

**Theoretical Claims:**

The proofs presented in the paper are logically structured and follow established techniques in runtime analysis. Further validation through sensitivity analysis, empirical studies, and exploration of alternative proof techniques would strengthen the theoretical claims and enhance their practical applicability.

---

> ### Author Rebuttal · Authors · 2025-04-01
>
> Your detailed comments are much appreciated, and we will revise the manuscript accordingly. Below, we address your questions.
>
> >1.How does the proposed runtime analysis scale with increasing problem complexity (e.g., higher-dimensional input spaces or larger numbers of classes)? Are there any limitations in extending the analysis to more complex scenarios?
>
> (1) We first explain the impact of increasing problem complexity by using larger numbers of classes (i.e., increasing $M$ in the analysis). In this case, the proposed runtime analysis scale will not be changed. This can be found from our theoretical results (Theorems 4.1 to 4.4) that reveal the relationship between runtime $T$ and classes number $M$, i.e., the expected runtime $\mathbb{E}[T]$ of (1+1)-ENAS grows polynomially with $M$. Next, we explain the impact of increasing problem complexity by using higher-dimensional input spaces. In this case, runtime analysis becomes more challenging due to that the fitness function is difficult to express in a mathematically tractable form. This difficulty arises from the fact that the mathematically formulated fitness function is derived by the hypervolume of the decision regions, while the higher-dimensional input spaces result in that the hypervolume of the higher-dimensional decision regions cannot be calculated from the geometric properties of the decision regions (i.e., higher-dimensional spaces make it difficult to provide a geometric interpretation of the neural network’s decision regions).
> (2) Yes, the main limitation is that the complex decision regions restrict the mathematical formulation of the fitness function. This, in turn, makes it difficult to directly assess the progress of the ENAS search process, which is a crucial step in the runtime analysis. We will revise to add more discussion. Thank you.
>
> >2.The fitness function assumes optimal parameter tuning for each architecture. How sensitive are the theoretical results to deviations from this assumption? Could the analysis be extended to account for suboptimal parameter tuning?
>
> (1) The theoretical results are robust to small deviations from optimal parameter tuning. Specifically, deviations from optimal parameter tuning will introduce noise into the fitness evaluation process, which could potentially alter the fitness landscape and disrupt the ranking of architectures' fitness values. However, if the fitness values maintain a high correlation with the true performance of the architectures, the impact on the ranking of candidate architectures is likely to be minimal, and thus unlikely to affect the search results in each round. In such cases, the theoretical results may not be notably affected. Overall, if the deviation is very small, the impact on the theoretical results is not significant.
> (2) To account for suboptimal tuning, the analysis could be extended via (1+$\epsilon$)-approximation or noisy optimization frameworks. We will discuss these directions in the revised manuscript and pursue them in future work. We appreciate your support as we advance this work.
>
> >3.The paper shows that one-bit and bit-wise mutations achieve similar performance. How do these findings influence the design of future ENAS algorithms?
>
> The design complexity of one-bit and bit-wise mutation–based ENAS algorithms differs: one-bit mutation is a simple and parameter-free operator that applies a single random change per step, while bit-wise mutation requires to tune the bit-flip probability (e.g., setting $p=1/n$ versus $p=0.5$ yields vastly different exploration-exploitation balances, where $n$ is the problem size). Our findings show that the simpler one-bit mutation can be prioritized in ENAS algorithm design. This is particularly effective for block/cell-based search spaces, where one-bit mutation operator could achieve significant changes in neural architecture. Furthermore, this aligns with practical ENAS methods like AE-CNN (Sun et al., 2019b), which utilize single-step mutation (functionally analogous to one-bit mutation) as their core search operator. We thank the reviewer for this comment and will revise to strengthen the discussion on mutation operator selection.
>
> >4.Can the theoretical framework be extended to analyze more complex ENAS algorithms? If so, what are the key challenges?
>
> Yes, the theoretical framework—including search space partitioning, optimization phase division, distance function definition, and individual transition probability analysis—can be extended to more complex ENAS algorithms. However, this extension faces key challenges: 1) how to calculate the probability of ENAS algorithm making progress in one step; 2) how to derive a tighter expectation of one-step progress (which is related with the progress derived by evolution operators); 3) how to select an appropriate runtime analysis tool based on the relationship between the distance at each generation (iteration) and the expected progress.

---

### Official Review · Reviewer_eb31 · 2025-03-14

**Overall Recommendation:** 4

**Summary:**

The paper delves into the runtime analysis of ENAS for multiclass classification，It introduces a new benchmark problem, MCC, designed to simulate multiclass classification tasks, and formulates a fitness function to evaluate neural architectures' performance on this problem. The authors also design a two-level search space, support detailed theoretical analysis. Empirical experiments further validate these theoretical results, highlighting the potential for simpler mutation strategies in ENAS design.

## Update after rebuttal

The authors' rebuttal looks great to me. My final recommendation is a clear accept.

**Claims And Evidence:**

In general, the claims made in the submission are backed by a combination of rigorous mathematical proofs and experimental results.

**Essential References Not Discussed:**

Yes, there are several related works that provide essential context for the key contributions of the paper.

**Experimental Designs Or Analyses:**

Yes, I reviewed the experimental designs and analyses presented in the submission.  The experiments focus on evaluating the performance of the (1+1)-ENAS algorithm with different mutation strategies (one-bit and bit-wise mutations) on the proposed MCC benchmark problem.

**Methods And Evaluation Criteria:**

The proposed methods and criteria are appropriate for the theoretical analysis.

**Other Comments Or Suggestions:**

The manuscript makes nice contributions to the theoretical understanding of ENAS for multiclass classification. The work of theoretical analysis is difficult and admire the author's efforts in this direction. More comments refer to weaknesses.

**Other Strengths And Weaknesses:**

**Pros as follows,**
1.The authors propose a multiclass classification benchmark problem (MCC) with a mathematically formulated fitness function. This is a significant contribution as it provides a standardized problem for analyzing ENAS algorithms.
2.The two-level search space (cells and blocks) is designed to be consistent with common ENAS settings, this design supports the theoretical analysis.
3.The theoretical results are supported by empirical studies, demonstrating the practical relevance of the findings.

**Cons as follows,**
1.The proposed MCC benchmark problem, while theoretically sound, is highly simplified and may not reflect the complexity of real-world multiclass classification tasks.
2.The runtime analysis focuses on the number of generations required to find an optimal solution but does not consider the computational resources (e.g., memory, processing power) needed to evaluate each candidate architecture.
3.The authors do not address how the proposed theoretical bounds scale with increasing problem complexity, which is crucial for understanding their practical feasibility.
4.The analysis assumes a relatively uniform and idealized search space, methods like dropout, batch normalization, or architectural constraints are not accounted for. These techniques can significantly influence the optimization landscape and runtime behavior of ENAS algorithms. Whether these techniques affect the optimization and runtime behavior of ENAS algorithms.

**Questions For Authors:**

Refer to weaknesses.

**Relation To Broader Scientific Literature:**

The paper makes several contributions that are well-aligned with the broader scientific literature.

**Theoretical Claims:**

I have checked the correctness of the proofs for the theoretical claims presented in the submission. As follows, Upper/Lower Bound on Expected Runtime for (1+1)-ENAS with One-bit/Bit-wise Mutation.

---

> ### Author Rebuttal · Authors · 2025-04-01
>
> Many thanks for your recognition and encouraging comments of our work. Below, we take the opportunity to respond to your concerns.
>
> >1.The proposed MCC benchmark problem, while theoretically sound, is highly simplified and may not reflect the complexity of real-world multiclass classification tasks.
>
> The MCC benchmark is designed to capture essential properties of real-world multiclass classification tasks, including both linearly and nonlinearly divisible decision regions (e.g., half-space region, unbounded/bounded polyhedra region). Many classification problems can be described as a disjoint union of these decision regions. While real-world tasks may involve additional complexities (e.g., noisy or more complex decision boundaries) that extend beyond the current MCC scope, the MCC benchmark provides a tractable yet representative framework for analyzing ENAS behaviors.  This simplified abstraction aligns with conventions in the theory community of evolutionary computation, where unimodal benchmark problems like OneMax and LeadingOnes similarly simplify real-world problems to analyze algorithmic behaviors (Droste et al., 2002; Doerr et al., 2008; Doerr & Goldberg, 2013; Witt, 2013). We will explicitly discuss the limitations of the MCC problem in the manuscript.
>
> >2.The runtime analysis focuses on the number of generations required to find an optimal solution but does not consider the computational resources (e.g., memory, processing power) needed to evaluate each candidate architecture.
>
> Our runtime analysis focuses on the expected number of fitness evaluations (i.e., the expected number of generations in this work) required to find an optimal solution, which concerns the ENAS search strategy and aligns with the theory community of evolutionary computation (Auger & Doerr, 2011; Neumann & Witt, 2010; Zhou et al., 2019; Doerr & Neumann, 2020). The computational cost (such as memory and processing power) per fitness evaluation is determined by the architecture training and validation process, which remains independent of the ENAS search strategy under analysis and consequently does not impact our runtime analysis. We will clarify this distinction in the manuscript.
>
> >3.The authors do not address how the proposed theoretical bounds scale with increasing problem complexity, which is crucial for understanding their practical feasibility.
>
> The problem complexity is determined by two parameters: $M$ (the number of classes) and $r$ (which affects the decision regions for each class). As proven in Theorems 4.1 to 4.4, the expected runtime $\mathbb{E}[T]$ of (1+1)-ENAS grows polynomially with $M$ and $r$. We will revise the manuscript to explicitly discuss this.
>
> >4.The analysis assumes a relatively uniform and idealized search space, methods like dropout, batch normalization, or architectural constraints are not accounted for. These techniques can significantly influence the optimization landscape and runtime behavior of ENAS algorithms. Whether these techniques affect the optimization and runtime behavior of ENAS algorithms.
>
> We fully agree that techniques like dropout, batch normalization, and architectural constraints can affect the optimization and runtime behavior of ENAS algorithms. Specifically, dropout and batch normalization can introduce noise into the fitness evaluation result, thereby effecting the optimization and runtime behavior of ENAS algorithms. Additionally, architectural constraints can reshape the search space, thereby altering the fitness landscape and consequently impacting the optimization and runtime behavior of ENAS algorithms. Current frameworks exclude these components due to unresolved challenges: (1) nonlinear interactions between architectural fitness and the noise introduced by dropout and batch normalization, complicating fitness modeling; (2) architecture constraints (e.g., parameter/resource limits) will introduce multi-objective optimization (e.g., balancing accuracy and efficiency). We will discuss these aspects and outline future directions in the Conclusion.

---

### Decision · Program_Chairs · 2025-05-01

**Decision:**

Accept (poster)

**Comment:**

This paper presents a theoretical runtime analysis of Evolutionary Neural Architecture Search (ENAS), specifically focusing on multiclass classification problems using a simple (1+1)-ENAS algorithm. The authors introduce a novel benchmark problem (MCC) designed for multiclass classification and a corresponding two-level search space representation. They derive theoretical upper and lower bounds (O(k^2 * n log n) and Ω(k^2 * n) respectively) on the expected runtime for (1+1)-ENAS using both one-bit and bit-wise mutation operators to find an optimal architecture within this framework. A key finding is that the simpler one-bit mutation achieves comparable theoretical performance to the more complex bit-wise mutation, a result supported by empirical validation. This work represents a significant step forward in the theoretical understanding of ENAS, extending prior analyses from binary to the more general multiclass setting.


The reviewers unanimously lean towards acceptance, with three recommending Accept (eb31, iu6X, 26fJ - with eb31 upgrading to Clear Accept post-rebuttal) and one recommending Weak Accept (dy7r). There is strong consensus on the paper's primary strengths.
The authors provided comprehensive and thoughtful rebuttals that effectively addressed most reviewer concerns.
All reviewers acknowledged the rebuttals, and they appeared largely satisfied with the responses, leading one reviewer to upgrade their score and others to maintain positive recommendations while acknowledging the remaining (mostly inherent theoretical) limitations.

This paper makes a solid and novel contribution to the theoretical understanding of ENAS. Establishing the first runtime analysis for the multiclass setting is a significant achievement. The proposed benchmark and search space provide a foundation for future theoretical work in this area. The finding regarding the effectiveness of simple one-bit mutation, supported by both theory and empirical results (including the additional ones in the rebuttal), offers a valuable insight that could potentially simplify ENAS design.

While the work operates within a necessarily simplified theoretical framework (a common characteristic of rigorous runtime analysis), the authors have convincingly argued for its relevance and have diligently addressed the reviewers' concerns regarding scope and assumptions. The strong consensus among reviewers, particularly after the clarifying rebuttals and additional experiments, supports acceptance. This work is a valuable addition to the ICML proceedings, pushing the boundary of theoretical understanding in an important area of automated machine learning.